

# Mode transitions in Northern Hemisphere Glaciation: Co-evolution of millennial and orbital variability in Quaternary climate

David A. Hodell[1] and James E.T. Channell[2]

[1]Godwin Laboratory for Palaeoclimate Research, Department of Earth Sciences, Downing Street, CB2 3EQ, UK
[2]Department of Geological Sciences, University of Florida, 241 Williamson Hall, POB 112120, Gainesville, FL 32611, USA

*Correspondence to*: David A. Hodell (dah73@cam.ac.uk)

**Abstract.** We present a 3.2-Myr record of stable isotopes and physical properties at IODP Site U1308 (re-occupation of DSDP Site 609) located within the ice-rafted detritus (IRD) belt of the North Atlantic. We compare the isotope and lithological proxies at Site U1308 with other North Atlantic records (e.g., Sites 982, 607/U1313 and U1304) to reconstruct the history of orbital and millennial-scale climate variability during the Quaternary. The Site U1308 record documents a progressive increase in the intensity of Northern Hemisphere glacial-interglacial cycles during the late Pliocene and Quaternary with mode transitions at ~2.7, 1.5, 0.9 and 0.65 Ma. These transitions mark times of change in the growth and stability of Northern Hemisphere ice sheets. They also coincide with increases in vertical carbon isotope gradients between the intermediate and deep ocean, suggesting changes in deep carbon storage and atmospheric $CO_2$. Orbital and millennial climate variability co-evolved during the Quaternary such that the trend towards larger ice sheets was accompanied by changes in the style, frequency and intensity of millennial-scale variability. This co-evolution may be important for explaining the observed patterns of Quaternary climate change.

**Key Words**: Integrated Ocean Drilling Program, Site U1308, North Atlantic, Quaternary, Paleoclimatology, Northern Hemisphere Glaciation

## 1 Introduction

For the last 2.7 Myr, Earth's climate has been characterized by the waxing and waning of large continental ice sheets in the Northern Hemisphere. Long sediment records from the North Atlantic basin recovered by IODP and its processor programs (DSDP and ODP) have provided a detailed history of Northern Hemisphere glaciation during the Pliocene and Quaternary. The intensification of



Northern Hemisphere glaciation began in the latest Pliocene and the intensity, shape, and duration of glacial-interglacial cycles changed during the Quaternary. The average climate state evolved towards generally colder conditions with larger ice sheets, and the spectral character of climate variability

shifted from a dominant period of 41 kyrs to a quasi-period between 80-120 kyrs. Glaciations were generally less intense, shorter in duration, and more symmetrical during the '41-kyr world' of the early Pleistocene (Maslin and Brierly, 2015). Across the Middle Pleistocene Transition (MPT), the glacial cycle lengthened, ice volume increased, and the shapes of marine isotopic stages assumed a more asymmetric, saw-tooth pattern during the '100-kyr world' of the late Pleistocene. The transition is often

viewed as a shift from a linear response of Earth's climate system to obliquity forcing prior to the MPT, to a more non-linear response afterwards, although the system response was likely more complicated (e.g., Ashkenazy and Tziperman, 2004).

The causes of these long-term patterns of Quaternary climate have been attributed to internal changes in climate response because orbital forcing did not change significantly over this time. Many

explanations have been invoked including:

1. Gradual $CO_2$ decline during the Quaternary resulting in long-term cooling and ice sheet growth (Raymo 1997);

2. Changes in global deep-ocean circulation resulting in modification of deep-ocean carbon storage capacity and heat distribution (Hodell and Venz-Curtis, 2006);

3. Glacial erosion and associated changes in ice-sheet dynamics (Pisias and Moore, 1981; Berger and Jansen, 1994; Clark and Pollard, 1998);

4. Sea ice switch mechanism as a result of the gradual cooling of the deep ocean during the Pleistocene (Tziperman and Gildor, 2003);

5. Ice-sheet behaviour as a multi-stable dynamical system with bifurcation points (Abe-

Ouchi et al., 2013, Ditlevsen, 2009);

6. Stochastic variability  (Meyers and Hinnov, 2010; Ditlevsen, 2009; Huybers, 2009)

None of these explanations are mutually exclusive and all could contribute by varying degrees to the observed patterns of Quaternary climate change.





Although it is accepted that orbitally-induced changes in insolation act as the pacemaker of the ice ages

(Hays et al., 1976), we still lack a complete understanding of what caused glacial-interglacial cycles

(Raymo and Huybers, 2008; Paillard, 2015).  This uncertainty is likely because the non-linear climate

system responds not only to longer-term external and internal forcing, but also to events (triggers) that

can result in major re-organization of the ocean-atmosphere system (Berger, 2013; Broecker and

Denton, 1989). Thus, it is important to understand how short-term (millennial) and long-term (orbital)

climate variability interact to produce the observed patterns of Quaternary climate change.

Observations of abrupt climate change in Greenland, beginning in the early 1990s (e.g., Dansgaard et

al., 1993), sparked a proliferation of studies of millennial-scale climate variability for the last glacial

cycle that is now being extended to older parts of the Quaternary.  The leading mechanism to explain

millennial-scale oscillations in the North Atlantic is fresh-water forcing of the strength of thermohaline

circulation (Broecker and Denton, 1989, among others), although other processes may also be involved

(Barker et al., 2015). The North Atlantic is one of the most climatically sensitive regions in the world

ocean because of its proximity to the North American, Greenland, and European ice sheets. Most of the

water stored in Northern Hemisphere ice sheets during Quaternary glaciations was discharged into the

Atlantic Ocean, either directly or indirectly via the Arctic (Fig. 1). The buildup of ice on Northern

Hemisphere continents can be thought of as a "capacitor" that stores freshwater on land during times of

ice growth and releases it to the ocean during times of ice decay as icebergs and meltwater (Bender,

2013). The volume, rate, and location of freshwater discharge to the North Atlantic Ocean relative to

the source areas of deepwater formation can have a strong impact on Earth's global climate.  As ice

sheets grew larger during glacial periods of the Quaternary, the freshwater capacitor became more

highly charged and the potential for a strong climate response upon discharge was enhanced.

Understanding the history of changes in the volume and location of ice build-up on Northern

Hemisphere continents and their impact on atmospheric and ocean circulation is important for

understanding Quaternary climate evolution.

Here we present a 3.2-Myr record of stable isotopes and physical properties at IODP Site U1308

(49°52.6661′N; 24°14.2875′W) (Fig. 1), which is located within the ice-rafted detritus (IRD) belt of the

North Atlantic (Ruddiman, 1977). Site U1308 represents the re-occupation of ODP Site 609, which has

played an important role in understanding Quaternary oribital and millennial climate change, including



the recognition of Heinrich events and correlation of millennial-scale climate variability between

marine sediment and Greenland Ice cores (Broecker et al., 1992; Bond et al., 1992, 1993, 1999;

McManus et al., 1994; Bond and Lotti, 1995). We integrate the isotope and lithological proxies from

Site U1308 with other North Atlantic records (e.g., Sites 982, 607/1313 and U1304) to elucidate the

patterns of orbital and millennial-scale variability in the subpolar North Atlantic during the Quaternary.

## 2 Methods

### 2.1 Composite section

Six holes were drilled at Site U1308 to ensure complete recovery of the stratigraphic section, and a

shipboard composite splice was constructed to 248 mcd (Expedition 303 Scientists, 2006a). The

composite for the upper 103 mcd was modified postcruise by Hodell et al. (2008) and we use the

revised mcd-scale as described by Channell et al. (2016).

### 2.2 Chronology

The integrated oxygen isotope and magnetostratigraphy of Site U1308 is described elsewhere for the

interval from 0 to 1.5 Ma (Hodell et al., 2008; Channell et al., 2008) and from 1.5 to 3.2 Ma (Channell

et al., 2016). The interval younger than 76 ka was dated using the age model of Obrochta et al. (2012,

2014) that is based on correlating variations in sediment lightness between Sites U1308 and 609, and

then transferring the Site 609 age model to U1308. The age model of Site 609 consists of recalibrated

radiocarbon dates and correlation of *Neogloboquadrina pachyderma* (sin) to Greenland ice-core $\delta^{18}O$

placed on the GICC05 age model. Beyond 76 ka, the age model was derived by correlating the benthic

$\delta^{18}O$ signal to the LR04 benthic oxygen isotope stack (Lisiecki and Raymo, 2005) assuming linear

sedimentation rates between tie-points. Modification to the oxygen isotope age model of Hodell et al.

(2008) was made near the transition of Marine Isotope Stage (MIS) 10 to MIS 9 (Termination IV)

where benthic foraminifera are very scarce and the original age model is inaccurate. In this interval,

analysis of $\delta^{18}O$ from planktonic foraminifera has permitted a refinement of the stratigraphy (Table 1).

The oxygen isotope record of Site U1308 indicates that the section is complete except for a short hiatus

that removed MIS G1 and G2 (Fig. 2), corresponding to the interval from ~2.6 to 2.65 Ma (Channell et

al., 2016).





**2.3 Stable isotopes**


Forminifera were picked from the >212-μm size fraction, and one to five individuals were used for analysis. Stable oxygen and carbon isotopes were measured on the benthic foraminifer *Cibicidoides wuellerstorfi* and/or *Cibicidoides kullenbergi*. The sample spacing for foraminifer stable isotopes was approximately every 2 cm for the upper 100 mcd (Hodell et al., 2008) and 5-10 cm for the 100-248 mcd interval of Site U1308. Foraminifer tests were soaked in ~15% $H_2O_2$ for 30 min to remove organic matter. The tests were then rinsed with methanol and sonically cleaned to remove fine-grained particles. The methanol was siphoned with a syringe, and samples were dried in an oven at 50°C for 24 hr. The foraminifer calcite was loaded into individual reaction vessels, and each sample was reacted with three drops of phosphoric acid (specific gravity = 1.92) using a Finnigan MAT Kiel III carbonate preparation device. Isotope ratios were measured online using a Finnigan MAT 252 mass spectrometer at the University of Florida. Analytical precision is estimated to be +/-0.08 ‰ for $\delta^{18}O$ and +/-0.03 ‰ for $\delta^{13}C$ by measuring eight standards (NBS-19) with each set of 38 samples.



Oxygen isotopes of bulk carbonate were measured using a ThermoScientific GasBench II, equipped with a CTC autosampler coupled to a DeltaV mass spectrometer (Spötl and Vennemann, 2003). Analytical precision is estimated to be ±0.1‰ for $\delta^{18}O$ by repeated analysis of the Carrara Marble standard. All isotope results are reported relative to Vienna Pee Dee Belemnite (VPDB). The sample spacing for bulk oxygen isotope measurements was approximately every 2 cm for the upper 100 mcd (Hodell et al., 2008) and 4 cm for the 100-248 mcd interval of Site U1308.


**2.4 Physical Properties**


Density and natural gamma radiation (NGR) were made on board the JOIDES Resolution during IODP Expedition 303 (Expedition 303 Scientists, Methods, 2006b). NGR was measured at a sample spacing of either 2.5 or 5 cm and density at 2.5-cm spacing. Because of the response curve of the NGR detectors is ~17 cm (full-width half maximum), the record is only suitable for studying orbital-scale variation. Volume susceptibility was measured at 1-cm intervals on u-channel samples (2x2x150 cm³ continuous subcores) using a susceptibility bridge designed for u-channel samples with a response function half-peak width of 3 cm (Thomas et al., 2003). The magnetic grain size parameter ($\kappa_{ARM}/\kappa$) was measured every 1-cm but each measurement is not independent of adjacent measurements owing to the ~4.5-cm width at half-height of the magnetometer response function (Channell et al., 2016).





The temporal resolution of physical properties measurements varies from 250 to 625 years assuming an

average sedimentation rate of 8 cm kyr$^{-1}$ for the entire record. Although interval sedimentation rates

vary considerably, the resolution of the physical properties should be sufficient to detect millennial

events, as demonstrated for Heinrich and other IRD events in the late Pleistocene (Hodell et al., 2008;

Channell et al., 2012).

**2.5 Time series analysis**

Traditional time series analysis was conducted using REDFIT and spectral peaks were evaluated

against a red-noise background from an AR1 process (Schulz and Mudelsee, 2002). To track the time-

varying amplitude of orbital and suborbital periods, we calculated the continuous wavelet transform

using the MatLab code of Grinsted et al. (2004). Time series for wavelet analysis were first Gaussian

interpolated to a fixed time increment of 1kyr with a 3-kyr window. The statistical significance of

wavelet power was tested relative to a red noise background power spectrum.

**3 Results**

**3.1 Stable isotopes**

**3.1.1 Oxygen isotopes**

We compare the benthic $\delta^{18}$O record of Site U1308 to Site 607 as the two sites are at similar water

depths (3900 m for U1308 and 3427 m for 607) and located within the same water mass today. The

benthic $\delta^{18}$O of Site U1308 is similar to Site 607 and the LR04 stack, but is of higher resolution (Fig.

3). Interglacial $\delta^{18}$O values are nearly the same for the two sites for the last 1.5 Myrs, and slightly

lower at Site 607 than U1308 prior to that time. Glacial $\delta^{18}$O values tend to be greater at Site U1308

than 607. In the latest Pliocene, glacial $\delta^{18}$O values show a progressive increase culminating in MIS

100, 98 and 96. Following MIS 96, the glacial stages were generally weaker from MIS 94 through 52

with the exception of MIS 82 (2.15 Ma), which was a strong glacial stage in the early Pleistocene

(Raymo et al., 1986). Beginning with MIS 52 (1.5 Ma), the amplitude of glacial-interglacial cycles

increased because of an increase in glacial $\delta^{18}$O and a decrease in interglacial $\delta^{18}$O values. From MIS

51 through 25, interglacial $\delta^{18}$O was close to the Holocene value of 2.8 ‰ except for MIS 31, 37 and

47 that were particularly strong interglacials. MIS 22 was strong compared to the preceding glacials but

not in comparison to MIS 16 and 12, which were the strongest glacial periods recorded for the last 3.2



Ma. The period from 790 to 480 ka was marked by "luke-warm" interglacials (MIS 19, 17, 15, 13) and interglacial stages became stronger again beginning with MIS 11.

Wavelet analysis of benthic $\delta^{18}O$ record indicates a strengthening of 41-kyr power at 2.6 Ma (Fig. 3).

In particular, the period between MIS 52 and 36 was marked by an exceptionally pure 41-kyr cycle. Longer periods in the range of 80-100-kyr began to appear at 900 ka and became dominant at 640 ka.

### 3.1.2 Carbon isotopes

The benthic $\delta^{13}C$ of Site U1308 is similar to Site 607 although the amplitude of the $\delta^{13}C$ signal is greater at Site U1308 than Site 607 for some intervals (Fig. 3). This may be a consequence of the

higher sample density of the Site U1308 record or its greater water depth. Our isotope measurements were made on a mixture of *C. wuellerstorfi* and *C. kullenbergi*. Although *C. kullenbergi* can possess a shallow infaunal habitat and lower $\delta^{13}C$ values compared to *C. wuellerstorfi* (Hodell et al., 2001), it is not obvious that the $\delta^{13}C$ of *C. kullenbergi* is any lower than that of *C. wuellerstorfi* at Site U1308.

The first large decrease in benthic $\delta^{13}C$ occurred in MIS G6 (2.71 Ma), but this event is not observed as

strongly in the Site 607 record (Fig. 3). Persistent decreases in glacial $\delta^{13}C$ values of <0 ‰ began with MIS 100 (2.52 Ma). Even during the early Pleistocene, glacial benthic $\delta^{13}C$ values are often as low as those of MIS 2. Beginning with MIS 52, glacial $\delta^{13}C$ values are consistently less than values in the last glacial, and are marked by strong 41-cyclicity between MIS 52 and 36. The Site 607 $\delta^{13}C$ record shows a step-like decrease in $\delta^{13}C$ beginning with MIS 52 (Raymo et al., 1990).

The lowest $\delta^{13}C$ values of the Site U1308 record occur in MIS 22 and 12. During the last million years, Site U1308 shows the same trends in benthic $\delta^{13}C$ as Site 607 where minimum glacial $\delta^{13}C$ values progressively increase from MIS 22 to 14 and again from MIS 12 to 2 (Raymo et al., 1990). The highest $\delta^{13}C$ values of the entire Site U1308 record occur in MIS 13.

Wavelet analysis of the benthic $\delta^{13}C$ record shows dominant 41-kyr power in the interval from 2500 to

640 ka followed by a shift to 100 kyr power after 640 ka. The interval from MIS 52 to 35 has an exceptionally pure 41-kyr cyclicity.



### 3.1.3 Bulk Carbonate $\delta^{18}O$

The $\delta^{18}O$ of bulk carbonate at Site U1308 reflects relative changes in the proportion of biogenic and

reworked (detrital or biogenic) carbonate. It is controlled by the input of reworked carbonate but also

by changes in the productivity of calcareous microfossils. If biogenic carbonate production is

supressed, then the $\delta^{18}O$ of bulk carbonate will reflect the isotopic composition of the remaining

carbonate, which consists of reworked (older) biogenic carbonate or detrital carbonate, delivered by

icebergs.

Hodell et al. (2008) showed that lows in bulk $\delta^{18}O$ are associated with IRD deposition although there

are two distinct types of events: one with low carbonate content and the other with elevated carbonate

concentrations. The low $\delta^{18}O$ low-carbonate events coincide with peaks in density and Si/Sr and are

associated with IRD that is rich in silicate minerals. Counts of IRD and foraminifera confirm that the

lows in bulk carbonate $\delta^{18}O$ correspond with lows in foraminiferal counts and peaks in lithic grains, at

least for the last several glacial cycles (Supplement Fig. S1; Obrochta et al., 2012, 2014). Low $\delta^{18}O$

high-carbonate events are limited to the last 650 ka and correspond to the lowest bulk $\delta^{18}O$ values (<-4

‰) and are associated with Ca/Sr peaks indicative of high concentrations of detrital carbonate in

Heinrich layers (Hodell et al., 2008; Hodell and Curtis, 2008).

A potential source of detrital carbonate is IRD originating from the Labrador Sea, which has $\delta^{18}O$

values averaging −5.6‰ ±1.5‰ (Hodell and Curtis, 2008). At Site U1308, the bulk carbonate $\delta^{18}O$

decreases to the Hudson Strait end member value for Heinrich events 1, 2, 4 and 5 (Hodell et al., 2008).

However, the $\delta^{18}O$ of bulk carbonate during Heinrich events 3 and 6 attains values of only -2‰, similar

to most of the events prior to 650 ka. Other possible sources of detrital carbonate include limestone and

chalk from the British-Irish ice sheet (Scourse et al, 2000; Peck et al., 2006) and reworked Cretaceous

and Paleogene chalk from Northwest Europe.

The bulk $\delta^{18}O$ at Site U1308 is marked by long-term, glacial-interglacial changes as well as abrupt

events (minima) during glacial periods that coincide with peaks in IRD (Fig. 4). We compare the bulk

carbonate $\delta^{18}O$ with the benthic foraminifer $\delta^{18}O$ signal because we expect both signals to increase



during glacial stages and decrease during interglacial periods. Divergence from this pattern indicates an

allochthonous source of carbonate to Site U1308 (Balsam and Williams, 1993; Hodell and Curtis,

2008). We subtract the bulk carbonate $\delta^{18}O$ from the foraminifer $\delta^{18}O$ record to reveal the differences

between the two records (Fig. 4 and Supplement S2).

Prior to 2.5 Ma, variations in bulk carbonate $\delta^{18}O$ closely follow the benthic $\delta^{18}O$ record with higher

values during glacials and lower values during interglacials (Supplement Fig. S3). Between 2.5 and 1.8

Ma, there are occasional events where bulk $\delta^{18}O$ is significantly less than foraminifer $\delta^{18}O$; for

example, during glacial MIS 100 and 98 with a particularly large difference in MIS 82. After ~1.8 Ma,

the bulk $\delta^{18}O$ record is often interrupted by brief minima during glacial periods. The first cluster of

low-$\delta^{18}O$ events occurred during MIS 64, 62 and 60 between ~1.8 and 1.7 ka (Fig. 4). After 1.5 Ma

(MIS 50), almost all glacial stages are marked by abrupt decreases in bulk carbonate $\delta^{18}O$ (Supplement

Fig. S4). In the interval from 1500 to 650 ka, most of the lows in bulk carbonate $\delta^{18}O$ are associated

with glacial inceptions and/or terminations (Supplement Fig. S4).

A pronounced change in the bulk carbonate $\delta^{18}O$ signal occurs at 650 ka during MIS 16 coincident

with the first occurrence of detrital carbonate layers (Heinrich events) derived from Hudson Strait

(Hodell et al., 2008). Bulk carbonate $\delta^{18}O$ values reach the detrital carbonate end-member values of -

5‰ and are associated with peaks in Ca/Sr (Fig. 4). After 650 ka, the bulk carbonate $\delta^{18}O$ minima

(some but not all are Heinrich events *sensu stricto*) occurred relatively late in the glacial cycle and

often on glacial terminations (Hodell et al., 2008) (Supplement Fig. S4).

Wavelet analysis of benthic-bulk $\delta^{18}O$ shows strong 41-kyr power between 1500 and 650 ka (Fig. 4).

Variability in the suborbital (millennial–scale) band also strengthens significantly after ~1.5 Ma. An

increase in the precession band (23-19 ka) occurs near 900 ka. In general, the strength of millennial

variability appears to be proportional to power in the precessional band. An increase in 100-kyr power

occurred at 450 ka. The power spectrum for bulk carbonate $\delta^{18}O$ contains significant power at ~100,

41, 21-19, 12.8, 9.7 and 8 kyr. (Fig. 5). The higher frequencies may reflect harmonics and/or

combination tones of the primary orbital cycles.



### 3.1.4 Natural gamma radiation

Natural gamma radiation is produced by the radioactive decay of K, Th, and U isotopes, which are contained in clays but can also originate from heavy minerals or lithic grains. Variations in NGR closely follow the benthic $\delta^{18}O$ record, increasing during glacial stages and decreasing during interglacial periods (Fig. 6). This pattern reflects a greater input of terrigenous sediment during glacial periods and increased carbonate productivity during interglacials. The first significant increase in NGR occurs in MIS 100 (2.52 Ma), consistent with increased delivery of terrigenous sediment to the subpolar North Atlantic at this time (Naafs et al., 2012; Lang et al., 2014).

Wavelet analysis indicates an increase in 41-kyr power beginning at 2.5 Ma and strengthening at 1.5 Ma. Spectra between MIS 52 to 36 are characterized by well-defined 41-kyr cycles, and 41-kyr power dominates until 640 ka when a cycle centered at power of 100-kyr emerges in the record (Fig. 6).

### 3.1.5 Density

In the late Pleistocene, peaks in density are well correlated with lithic grains per gram at Site U1308 (Obrochta et al., 2014). In addition, density varies on glacial-interglacial cycles and increases slowly down-core owing to sediment compaction. The first large density peak occurs just below the hiatus that removed MIS G1 and G2. Strong peaks are recorded in MIS 100 and 98 (Fig. 6), consistent with the widespread delivery of ice-rafted detritus to the subpolar North Atlantic at this time (Shackleton al., 1984; Kleiven et al., 2002; Bailey et al., 2010, 2012; Bolton et al., 2010). Density peaks are associated with certain glacial stages including MIS 86, 82-78, 74-72, 64-62-60, 52, 46, 40, 36, 34, 28-24-22, and 18. The most outstanding feature of the density record are the large peaks beginning at 640 ka in MIS 16 that are associated with Heinrich layers (Hodell et al., 2008; Channell et al., 2012).

Wavelet analysis reveals that density has generally weak power in the orbital band (Fig. 6). Stronger power in the millennial band begins at ~1.8 Ma (MIS 64).

### 3.1.6 Magnetic Susceptibility and Grain Size

Magnetic susceptibility reflects the total concentration of magnetic minerals, but is usually dominated by the concentration of magnetite owing to its high intrinsic susceptibility, whereas $\kappa_{ARM}/\kappa$ is a grain size proxy for magnetite. In general, glacial isotopic stages are associated with higher magnetic susceptibility and a tendency towards coarser magnetic grain sizes (i.e., lower values of $\kappa_{ARM}/\kappa$) (Fig.





7). Peaks in IRD abundance are similarly marked by abrupt peaks in magnetic susceptibility and coarsening of the magnetic grain size parameter (Channell and Hodell, 2013). IRD transported by icebergs or sea-ice from volcanic source areas (e.g., Iceland) are expected to have a disproportionately large effect on magnetic susceptibility.

Magnetic susceptibility begins to increase during glacial periods at 2.4 Ma with especially high values in MIS 82. Magnetic susceptibility is also high in glacial periods between MIS 74 and 58 between 2 and 1.65 Ma. From MIS 50 (1500 ka) to 16 (650 ka), there is a regular pattern of glacial increases and interglacial decreases in magnetic susceptibility. From 650 ka onward, the Heinrich events are marked by large peaks in magnetic susceptibility. The main feature of the wavelet of magnetic susceptibility is the activation of the millennial band after ~2 Ma.

For $\kappa_{ARM}/\kappa$, MIS 100, 98 and 96 show a muted increase in magnetite grain size consistent with IRD delivery (Fig. 7). MIS 82 is associated with a strong coarsening of magnetic grain size as are MIS 64, 62, 40 and 34. A marked coarsening of magnetic grain size occurs during Heinrich events from 650 ka onwards. Beginning with MIS 19 at 750 ka, there is a distinct decrease in magnetite grain size during interglacials.

## 4 Discussion

The discussion is organized around major climate transitions identified at ~2.7, 1.5, 0.9 and 0.65 Ma. The timing of these transitions is not exact and can vary significantly depending upon the proxy considered because of leads and lags of different components of the ocean-atmosphere system. Estimates for the intensification of Northern Hemisphere glaciation in the late Pliocene range from 3.5 to 2.4 Ma (Kleiven et al., 2002). A lesser known transition occurred in the middle Pleistocene near 1.6 to 1.5 Ma (Rutherford and D'Hondt , 2000; Hodell and Venz, 2006; Lisiecki, 2014). The onset and end of the Middle Pleistocene Transition have been placed between 1250 ka and 650 ka, respectively (Clark et al., 2006). These transitions represent times of fundamental re-organization of the climate system, and may mark bifurcation points in a dynamic climate system characterized by multiple stable states (Ditlevsen, 2009).





### 4.1 Intensification of Northern Hemisphere Glaciation (NHG)

The intensification of Northern Hemisphere glaciation in the latest Pliocene is well documented in

North Atlantic sediments. New data from this study are consistent with previous findings at Site

U1308 for MIS G8 through 100 (Bailey et al., 2010, 2012). At Site U1308, benthic $\delta^{18}O$ exceeds 3.5 ‰

for the first time in MIS G6 (~2.72 Ma), and corresponds with a pronounced decrease in benthic $\delta^{13}C$

(Fig. 3). A brief hiatus at Site U1308 removed the record from ~2.6 to 2.65 Ma including MIS G1 and

G2 (Fig. 2). The record of natural gamma radiation indicates a significant increase in terrigenous input

relative to biogenic carbonate during glacial periods beginning with MIS 100 (2.52 Ma) accompanied

by the onset of a distinct 41-kyr cycle (Fig. 4). Increased terrigenous input during glacials beginning at

MIS 100 could have been derived from both dust and IRD (Naafs et al., 2012; Lang et al., 2014).

Density peaks occur in MIS 100 (2.52 Ma) and 98 (2.48 Ma) indicate the occurrence of IRD (Fig. 5).

Geochemical provenance studies of IRD carried out for MIS 100 at Site U1308 suggest multiple

sources (Bailey et al., 2010, 2012). The early delivery of IRD was mostly from Archean (Greenland)

sources, whereas the source shifts to early Proterozoic and Caledonian-age rocks (probably from

Scandinavia and North America) during the full glacial conditions of MIS 100 (Bailey et al., 2012).

The $\delta^{18}O$ of bulk carbonate decreases slightly during MIS 100 and 98, but these events are weak in

comparison to later glacial periods (Fig. 5). MIS 100-98-96 were strong glacial periods when benthic

$\delta^{18}O$ values at Site U1308 approach those of MIS 4 (Fig. 3). Land-based evidence suggests that the

Laurentide Ice Sheet advanced to 39°N at 2.4 Ma, with the next similarly extensive advance occurring

at ~1.3 Ma (Balco and Rovey, 2010).

Glacial benthic $\delta^{13}C$ values in the deep North Atlantic decreased at 2.7 Ma (Fig. 3), which has been

interpreted as indicating a decrease of Northern Component Water (NCW) (Raymo et al., 1992). In the

Southern Ocean, the carbon isotope gradient between intermediate and deep-water ($\Delta^{13}C_{ID}$) increased at

~2.75 (MIS G6) marking the development a chemical divide in the Atlantic Ocean between well-

ventilated intermediate water and more poorly-ventilated deep water. Hodell and Venz-Curtis (2006)

speculated that this change may signify increased carbon storage in the deep-sea and hence a decrease

in atmospheric $pCO_2$. This prediction appears to be supported by recent paleo-$CO_2$ reconstructions

(Martinez-Boti et al., 2015; Bartoli et al., 2011) and modelling studies (Lunt et al., 2008; Willeit et al.,

2015).



The strong glacial triplet of MIS 100-98-96 represented a temporary intensification of glaciation, likely related to a particular set of orbital conditions that included an exceptionally low eccentricity and

dampened precession cycle (Maslin et al., 1998). Generally weaker glacials followed from MIS 94 to MIS 52 (1.54 Ma) with the exception of MIS 82 (2.15 Ma), which was a strong glacial stage in the early Pleistocene that coincided with unusually low obliquity. Raymo et al. (1986) detected no IRD between MIS 89 and 95 at Sites 607 and 609, whereas IRD was detected in MIS 86 and 88 at Site 609 but not at Site 607.

Between MIS 94 and 52, glacial benthic $\delta^{18}$O values rarely exceed 4 ‰ with the notable exception of MIS 82. Ice sheet conditions during glacial periods may have been similar to MIS 5b when modelling results suggest the Laurentide Ice Sheet was reduced in size with separate Quebec and Keewatin domes (Supplement Fig. S6a; Kleman et al., 2013). The Scandanavian and Barents-Kara ice domes were well developed in MIS 5b but did not extend into central Europe or the British Isles (Kleman et al., 2013).

Ice volume was comparable on North America and Europe at this time, and the ice-sheets responded dominantly to obliquity forcing during this period (Abe-Ouchi et al., 2013).

Benthic $\delta^{13}$C values during the early Pleistocene period were often as low as those in the LGM, suggesting a shoaling of NCW, increased Southern Component Water (SCW) influence, and/or decreased ventilation (Fig. 3). Although glacial ice volume was substantially reduced during the early

Pleistocene relative to the LGM, the impact on deep-water circulation may have been substantial. The Eurasian ice sheets may have had a disproportionately larger effect on deep-water formation during the early Pleistocene because of their proximity to the Norwegian-Greenland Sea (Fig. 1). Even the relatively small North American ice sheets in the early Pleistocene may have had an effect on deep-water circulation because of their proximity to the Labrador Sea.

The magnitude and spatial extent of millennial-scale variability during the late Pliocene and earliest Pleistocene is uncertain. At Site 984 (61°25.507'N, 24°04.939'W), Bartoli et al. (2006) reported significant millennial-scale variability during glacial stages following the intensification of NHG (2.9–2.8 Ma). In contrast, Bolton et al. (2010) found no significant amplification of millennial variability during the glacials of MIS 100, 98, and 96 at Site U1313 (41°0.0679′N; 32°57.4386′W). They further

suggested that the threshold for amplification of millennial variability was not crossed during the late





Pliocene and likely not until the Mid-Pleistocene Transition. Alternatively, the occurrence of strong

millennial variability, as in the late Pleistocene, may have been limited to higher latitudes than Site

U1308/U1313 during the late Pliocene-early Pleistocene.

Magnetic susceptibility begins to show an increase in power in the millennial band after 2 Ma (Fig. 7).

This change may be related to glaciation in Iceland where glaciers did not reach sea level until ~2.0 Ma

(Einarsson and Albertsson, 1988; Geirsdóttir, 2004). Bulk $\delta^{18}$O and density values imply a series of

three IRD events associated with MIS 64, 62 and 60 (Fig. 5). This appears to coincide with an increase

in magnetic susceptibility at Site 984, south of Iceland, indicating increased delivery of volcanic IRD

by icebergs and/or sea-ice in glacial periods beginning at 1.8 Ma with MIS 64 (Channell et al., 2002).

McIntyre et al. (2001) examined millennial variability at Site 983, south of Iceland, for two periods in

the early Pleistocene (1.86 -1.93 Ma and 1 .75-1.83 Ma), including MIS 64 and 70. They found clear

evidence for millennial IRD events that recurred approximately every ~2-5 kyr in these two glacial

stages

### 4.2 The 1.5 Ma transition

The climate transition at ~1.5 Ma represented a fundamental change in the mode of glacial-interglacial

climate cycles, yet it has received relatively little attention. MIS 52 (~1.54 Ma) marked an important

change at Site U1308 as millennial-scale variability increased in the mid-latitude North Atlantic. The

increased frequency of bulk carbonate $\delta^{18}$O events at 1.5 Ma signals the persistent delivery of detrital

carbonate to Site U1308 during glacials from MIS 50 onwards (Fig. 5).

We suggest the lows in the $\delta^{18}$O of bulk carbonate between 1.54 and 0.64 Ma were similar to the non-

Heinrich IRD layers in the late Pleistocene, which are marked by peaks in lithic grains and low

foraminifer abundance. Many of the bulk $\delta^{18}$O lows between 1.5 and 0.65 Ma are associated with

glacial inceptions and some with terminations (Supplement Fig. S4). These IRD events likely reflect

climate-driven changes in the mass balance of ice sheets as a result of advance and retreat of the

grounding line at multiple locations in the circum-Atlantic region (Marshall and Koutnik, 2006).

After 1.54 Ma (MIS 52), benthic $\delta^{18}$O consistently exceeded 4 ‰ during glacial periods and ice sheet

conditions may have been similar to MIS 4 when the Laurentide Ice Sheet expanded over Hudson



Strait and a high saddle existed connecting the Quebec and Keewatin domes (Supplement Fig. S6b; Kleman et al., 2013). Full expansion of the Quebec Dome, to an extent comparable to the LGM,

occurred along the eastern margin of North America during MIS 4 (Kleman et al., 2013), and an ice stream existed in Hudson Strait, thereby supplying detrital carbonate to the North Atlantic during glacial periods although not by large dynamic surges typical of late Pleistocene Heinrich events. Ice volume was about twice as great in North America compared to Eurasia, and in the interval from MIS 52 to 36, benthic $\delta^{18}O$ was marked by an exceptionally well-defined 41-kyr cycle.

The size and position of the North American ice sheets have a strong downstream effect over the North Atlantic (Roberts et al., 2014; Ullman et al., 2014), including the position of the winter and summer sea ice limits (Lofverstrom et al., 2014; Supplement Fig. S7). In turn, iceberg drift (and melting) is affected by atmospheric circulation and tends to follow the zero curl of the wind stress. Sea surface temperature (SST) at Site U1313 shows a strong cooling trend beginning at 1.5-1.6 Ma (Fig. 8)

(Lawrence et al., 2010; Naafs et al., 2012). During glacial stages beginning at 1.5 Ma, SST in the subpolar North Atlantic cooled and zonal SST gradients between Sites 982 (57.5°N) and 1313 (43°N) decreased (Fig. 8). The Polar Front moved south and winter sea ice extended to the position of Site U1308 during glacial periods. It's unclear whether the increase in glacial IRD after 1.5 Ma represents an increase in production of icebergs, or transport and survivability of icebergs out to Site U1308. At

Site U1313 to the south, IRD doesn't begin to greatly increase until ~900ka (Fig. 9), indicating that changes in IRD abundance are diachronous with latitude, occurring earlier in the north and later in the south.

On the basis of the benthic $\delta^{13}C$ of Site 607, Raymo et al. (1990) suggested a significant decrease in the production of North Atlantic Deep Water (NADW) after 1.5 Ma. A change in deep-water circulation at

1.6-1.4 Ma is also supported by Nd isotope studies in the North Atlantic (Khélifi and Frank, 2013), which imply that the overflow of deep waters from the Nordic Seas strongly decreased at this time. Hodell and Venz-Curtis [2006] identified ~1.55 Ma as an important time when the intermediate-deep ($\Delta^{13}C_{ID}$) gradient increased in the glacial South Atlantic Ocean, indicating either increased glacial suppression of NADW and/or reduced ventilation of southern sourced water. Lisiecki (2014) also

reported changes in Atlantic circulation at 1.5–1.6 Ma as indicated by the appearance of glacial $\delta^{13}C$ gradients between the intermediate and middle deep Atlantic $\delta^{13}C$ stacks. Hodell and Venz-Curtis



(2006) speculated that the increase in $\Delta^{13}C_{ID}$ at 1.55 Ma resulted in increased carbon storage in the deep ocean during glacials, and therefore may have been accompanied by lowering of atmospheric $CO_2$. On the other hand, boron-isotope-based $CO_2$ reconstructions do not support a major decrease in

$CO_2$ at 1.5 Ma, although the existing data have low resolution and additional studies are required (Hönisch et al., 2009).

At 1.55 Ma, climate in the polar regions of the North and South Atlantic became synchronized such that both the Arctic and Antarctic Polar Fronts were marked by glacial-interglacial migrations at a regular pacing of 41-kyr (Hodell and Venz, 1992). Synchronization of Northern and Southern

Hemispheres from ~1.55 Ma onward may indicate that changes in deep-sea carbon storage and $CO_2$ variations began to play an increasingly important role in glacial-interglacial climate change. This was also a time when dust and iron accumulation began to increase in the subantarctic South Atlantic, suggesting increased iron fertilization and $CO_2$ drawdown during glacial periods (Martinez-Garcia et al., 2011).

MIS 50 also marks a time of increased variability in the millennial band of bulk carbonate $\delta^{18}O$ and density at Site U1308 (Fig. 5), indicating an increased occurrence of IRD events in the subpolar North Atlantic. Millennial-scale variability was particularly strong between 1.5 and 0.65 Ma at Site U1308. McManus et al. (1999) suggested that the amplitude and frequency of variability in ice-rafting and sea-surface temperature proxies increases when ice volume is within a critical window defined by benthic

$\delta^{18}O$ values between 3.5 and 4.5 ‰. From 1.5 to 0.65 Ma, the climate system crossed the 3.5 ‰ threshold more often during the 41-kyr world than it did during the 100-kyr world and never exceeded the upper threshold of 4.5 ‰; thus, the climate system spent more time in the "DO window" (Sima et al., 2004). The benthic $\delta^{18}O$ threshold may have also been somewhat lower than 3.5 ‰ in the early Pleistocene, if ice sheets flowed more readily than their late Pleistocene counterparts (Raymo et al.,

1998; Bailey et al., 2010).

The Site U1308 record suggests that millennial variability was more active from 1.5 Ma, albeit without the glacial dynamics associated with Heinrich events (Hodell et al., 2008). This is supported by results from Site U1385 that demonstrate that millennial variability was a persistent feature on the Iberian margin since 1.5 Ma (Hodell et al., 2015). For example, Birner et al. (2016) showed that the magnitude





and pacing of millennial variability during MIS 38 and 40 at Site U1385 was similar to D-O cycles of

MIS 3. At Site 983, Raymo et al. (1998) demonstrated millennial-scale variability in MIS 40 and 44 on

the basis of proxies of iceberg discharge and deep-water chemistry.

Many of the inferred IRD peaks correspond with low benthic $\delta^{13}$C values at Site U1308 during the past

1.5 Myr, suggesting a link between iceberg discharge and weakening of thermohaline circulation

(Hodell et al., 2008). Although Site U1308 represents a single site at 50°N, extension of sea ice to the

mid-latitude North Atlantic and changes in Atlantic Meridional Overturning Circulation (AMOC) have

been shown to have widespread climate implications (Chiang and Bitz, 2005).

**4.3 The 900 ka event**

Elderfield et al. (2012) deconvolved the benthic $\delta^{18}$O record at Site 1123 in the SW Pacific into its

temperature and $\delta^{18}$O$_{water}$ components by tandem measurement of Mg/Ca and $\delta^{18}$O in benthic

foraminifera.  They inferred a step-like increase in glacial ice volume at ~900ka. Berger et al. (1993,

1994) also found a step-like increase in $\delta^{18}$O values beginning at 900 ka during glacials from shallow-

dwelling planktonic foraminifera on the Ontong-Java Plateau in the western Pacific warm pool (Fig.

10). These records provide strong evidence that the increase in glacial ice volume across the Middle

Pleistocene Transition (MPT) was not gradual but rather occurred abruptly at 900 ka during MIS 22-

23-24 (Berger, 1993, 1994; Berger and Jansen, 1994; Elderfield et al., 2012).

MIS 22-23-24 is often considered to be the first 100-kyr cycle because of its similarity with MIS 1-2-3.

The MIS 21-24 interval constitutes a lengthy glacial because MIS 23 was a weak interglacial and the

MIS 24/23 transition is not considered to be a termination (much like the MIS 4/3 transition).

Elderfield et al. (2012) proposed that the ice volume increase during MIS 22-23-24 may have occurred

on Antarctica in response to weak insolation during MIS 23 that suppressed substantial melting of the

ice formed in MIS 24.  Sea level lowering during MIS22-23-24 may have also permitted the advance of

marine-based ice sheets onto the continental shelves in the Northern Hemisphere.  The onset of

widespread glaciation in northern Europe appears to have occurred during MIS 22 in the circum-Baltic

region and in Alpine Europe, at the same time as the expansion of widespread lowland glaciation in

North America (Head and Gibbard, 2015). Berger and Jansen (1994) suggested the Svalbard–Barents

Sea and Kara Sea ice sheets advanced over the shelf areas after 1 Ma.  An over-consolidated section at



Site 910 on the Yermak Plateau further supports the grounding of a marine-based ice sheets from

Svalbard, and perhaps the Barents Sea, prior to 660 ka (Flower, 1997).

We observe only minor changes at Site U1308 in physical properties or $\delta^{18}$O at 0.9 Ma when global ice

volume increased. Instead, most physical properties show a transition at 650 ka coincident with the

deposition of the first Heinrich layer derived from Hudson Strait (see next section). At Site U1308,

there are two decreases in bulk carbonate $\delta^{18}$O during MIS 22-23-24 but neither of these events are

associated with peaks in detrital carbonate (e.g., increases Ca/Sr) indicating Heinrch layers.

In contrast, Site 982 on the Rockall Plateau (57°30.992'N, 15°52.001'W; 1145 m water depth) shows

three prominent decreases in bulk carbonate $\delta^{18}$O during MIS 22-23-24 (Fig. 11). These $\delta^{18}$O lows are

associated with decreases in % carbonate and increases in % IRD. The decreases in bulk $\delta^{18}$O during

MIS 22-24 indicate the delivery of reworked carbonate to Site 982 at 900 ka. At Site 980/981 in

Rockall Trough, increases in reworked nannofossil taxa coincide with IRD peaks (Marino et al., 2011).

The nannofossils are mostly of Cretaceous age, from the Campanian-Maastrichtian stages, and derived

from the Norwegian shelf and/or the northern North Sea–Denmark area (Marino et al., 2011).

Reworked specimens of mostly Cretaceous nannofossils also reach a maximum within the 900-700 ka

interval in cores from ODP Leg 104 off Norway (Henrich and Bauman, 1994). We suggest that the

minima in bulk carbonate during MIS 22-23-24 at Site 982 were related to expansion of the Eurasian

Ice Sheet as sea-level lowering permitted an increase in ice mass on the continental shelves of

Scandanavia and the Barents Sea (Berger and Jansen, 1994). The 900 ka event may not have been

associated with a major change in the stability of the Laurentide Ice Sheet as there is no evidence for a

Heinrich event associated with MIS 22, although the event probably correlates with ice-volume

changes in Antarctica and Eurasia, as well as expansion of lowland glaciation in North America.

As noted by Raymo et al. (1997), global benthic $\delta^{13}$C records show a pronounced transient decrease at

~0.9 Ma (Fig. 3), representing a perturbation of mean ocean carbon chemistry. Elderfield et al. (2012)

attributed this event to changes in deep-water circulation and erosion of organic carbon due to exposure

of slope and upper shelf deposits when sea level dropped to -120 m for the first time. Expansion of

southern sourced water into the deep Atlantic is supported by carbon and Nd isotopes that reflect



weaker NADW export to the Southern Ocean beginning about 0.9 Ma (Venz and Hodell, 2002; Pena
        and Goldstein, 2014).

        Raymo et al. (1997) noted that over the last 1.5 Ma, the intensity of $\delta^{13}C$ decreases during glaciations
        does not directly match the magnitude of $\delta^{18}O$ increase. For example, there is a progressive increase in
        glacial benthic $\delta^{13}C$ values from MIS 22 to 14 that is repeated again between MIS 12 and 2 (Fig. 3),

which is not seen in the benthic $\delta^{18}O$ record. These trends match the magnitude of the terminal IRD
        peaks found at Site 982 and coincide with transient decreases in benthic $\delta^{13}C$ at glacial terminations
        (Venz et al., 1999). These events indicate iceberg melting that lowered surface water salinity and
        thereby reduced Glacial North Atlantic Intermediate water, resulting in decreased ventilation of the
        mid-depth North Atlantic.

We suggest that the size and position of the Eurasian ice sheet was a critical factor responsible for the
        decoupling of benthic $\delta^{13}C$ and $\delta^{18}O$ because of the proximity of the ice sheet to source areas of deep-
        water formation in the Norwegian-Greenland Sea and subpolar North Atlantic (Fig. 1). Although the
        Eurasian ice sheet had a relatively small impact on the global $\delta^{18}O$ of seawater, it may have had a
        disproportional effect on deep-water formation and benthic $\delta^{13}C$, causing a decoupling of the response

of benthic $\delta^{18}O$ and $\delta^{13}C$ signals.

**4.4 The 640 ka transition and the emergence of a quasi-periodic 100-ka cycle**

        Undoubtedly the time of greatest change in stable isotopes and physical properties at Site U1308
        occurred in MIS 16 at ~650 ka. This change was discussed by Hodell et al. (2008) and marked a
        pronounced shift in the style and intensity of glacial-interglacial cycles and IRD delivery to the

subpolar North Atlantic.  Detrital carbonate layers (Heinrich events) first appeared at Site U1308,
        recorded by large peaks in density and magnetic susceptibility and minima in bulk $\delta^{18}O$ and $\kappa_{ARM}/\kappa$
        (Figs. 5 and 7; Hodell et al., 2008; Channell et al., 2012).  The results from Site U1308 are supported
        by results from Site U1313 to the south (Fig. 1) where organic biomarkers indicative of petrogenic
        compounds derived from Hudson Strait, and an increase in dolomite/calcite ratios, are observed from

0.65 Ma (Fig. 9; Naafs et al., 2013).



As the glacial cycle lengthened, the quasi-periodic cycle of 100 kyrs became firmly established at 650 ka (Figs. 3 and 4). Maslin and Ridgewell (2005) have coined the phrase "eccentricity myth" to describe the incorrect attribution of the increase in 80-120-kyr power to orbital eccentricity. Instead, the saw-toothed climate cycles are defined by every four or five precession cycles (Raymo, 1997) or

every two or three obliquity cycles (Huybers and Wunsch, 2005; Huybers, 2009), or some combination thereof. Eccentricity modulates the amplitude of the precession cycle and thus may play a role in pacing terminations, but the direct insolation changes resulting from eccentricity are too small to drive the 100-kyr cycle. After 0.9 Ma, the duration of glacial cycles appears to be quantized as multiples of either precession or obliquity (i.e,. 80, 100, or 120 kyrs).

MIS 16 marked the highest benthic $\delta^{18}O$ values of the entire record, suggesting very cold bottom water temperatures and/or increased ice volume (Fig. 3). MIS 16 and 12 have been referred to as superglacials, indicating continental ice volumes greater than during the LGM. During full glacial periods beginning with MIS 16, it is likely the domes of the North American Ice Sheet coalesced to form a massive unified North American Ice Sheet extending from coast to coast (Bintanja and van der

Wal, 2008; Supplement Fig. S6c). Hudson Bay was covered by a thick ice sheet, and ice streams may have terminated at peripheral ice shelves along the eastern Canadian seaboard (Hulbe et al., 2004). Widespread lowland glaciation was established in northern Europe in MIS 16, together with important expansion of lowland glaciation in North America (Head and Gibbard, 2015). The coalescing of the ice domes in North American permitted the ice sheet to survive subsequent insolation maxima and skip

precession or obliquity cycles, thereby lengthening the glacial cycle and transferring power from the precessional and obliquity band to the 80-120 kyr band.

The growth of very large ice-sheets also involved fundamental changes in the dynamics of the Laurentide Ice Sheet by introducing instabilities related to processes such as basal melting, isostatic subsidence, and/or drawdown during marine invasion. In addition, the topographic and albedo changes induced by very large ice sheets may lead to non-linear response of the climate system (Kleman et al.,

2013). Large ice sheets charge the freshwater capacitor, permitting the climate system to respond quickly and strongly when the capacitor discharged. The magnitude of the response was governed by the volume, rate, and location of freshwater addition relative to areas of active deep-water formation (Fig. 1).



IRD events were more frequent, and detrital carbonate layers absent from the record, prior to 650 ka

        (Fig. 5). This trend is also evident in the NGR record of Site U1304 that shows greater millennial

        variability prior to 650 ka than afterwards (Fig. 12). As argued previously, millennial variability was

        more frequent in the 41-kyr world because the climate system spent more time in an intermediate ice

        volume state and within the "DO window" (Sima et al., 2004). The decrease in the frequency of IRD

events during the past 650 kyrs may have also been related to formation of ice shelves, which act as a

        filters of IRD (Alley et al., 2005). Icebergs derived from ice shelves tend to have low debris

        concentrations because much of the debris is lost by basal melting before the iceberg is calved (Drewry

        and Cooper, 1981). IRD events after 650 ka occurred less frequently but were greater in magnitude

        than those prior to MIS 16 (Hodell et al., 2008). Ice-sheet dynamics played a crucial role in the

emergence of the saw-tooth pattern of glacial-interglacial cycles in the late Pleistocene. It is not

        coincidental that Heinrich events (especially terminal Heinrich events) first appeared when the quasi-

        100-kyr cycle became firmly established as indicated by time series analysis (Mudelsee and Stattegger,

        1997) (Fig. 3). The development of massive ice sheets during glacial periods beginning at 0.65 Ma

        (MIS 16) introduced a new type of millennial variability related to episodes of internal dynamical

instability of the LIS in the region of Hudson Strait (Hodell et al., 2008).

        **4.5 Co-evolution of millennial and orbital climate variability**

        The general trend of climate during the Plio-Pleistocene has been one of progressive buildup of larger

        ice sheets on Northern Hemisphere continents and increased amplitude of glacial-interglacial cycles.

        Although it is somewhat arbitrary to identify precise change points in the benthic $\delta^{18}O$ record (Fig. 13),

we propose mode transitions in climate evolution during intensification of NHG at approximately 2.7,

        1.5, 0.9 and 0.65 Ma. The timing of some of these mode changes is supported by Bayseian change-

        point analysis of the LR04 stack (Ruggieri, 2012).

        The benthic-bulk carbonate $\delta^{18}O$ difference at Site U1308 also shows a trend of increased variability

        through the Quaternary (Fig. 13A), representing a progressive increase in IRD delivery to the mid-

latitudes of the North Atlantic. There is a clear increase in IRD delivery to Site U1308 beginning about

        1.5 Ma that coincides with greater amplitude of the benthic $\delta^{18}O$ signal. At 0.65 Ma, Heinrich Events

        appeared for the first time introducing a new style of glacial dynamic related to exceptionally large ice

        sheets and long-duration glacial periods (Hodell et al., 2008).



As ice sheets reached the coast, they began to interact with the ocean and affect ocean circulation in
areas of deep-water formation such as the Nordic and Labrador Seas. As ice sheets continued to grow
in size, they stored greater volumes of fresh water on land close to the sources of deep-water formation
in the North Atlantic (Fig. 1), increasing the potential of triggering a large climate response. Thus, the
magnitude of millennial-scale climate variability may be related to ice sheet size (McManus et al.,
1999; Weirauch et al., 2008).

We suggest that the variability on orbital and millennial scales may be intrinsically linked, and
therefore co-evolved during the Quaternary.  There's much discussion about the role that millennial
variability plays in glacial terminations and inceptions. It is uncertain whether suborbital variability is
merely a symptomatic feature of glacial climate or, alternatively, plays a more active role in the
inception and/or termination of glacial cycles. For example, strong terminal millennial events may lead
to $CO_2$ degassing in the Southern Ocean through a bipolar seesaw mechanism (Denton et al., 2010;
Skinner et al., 2012), thereby hastening deglaciation. Millennial variability may also play a role in
glacial ice-sheet build-up by diminishing melting and stabilizing ice sheets during insolation maxima
(Timmermann et al., 2010).

Millennial variability acts as an 'agitator' of the climate system and can trigger transitions as
bifurcation points are approached in a multi-stable dynamical system or act as an activator in an
excitable system (Crucifix, 2012). Millennial variability can be considered a form of "noise" on orbital
time scales even though it may be derived from deterministic processes. Noise intensity has been found
to be important for stochastic resonance in systems containing a sub-threshold pacemaker (Perk and
Gosak, 2008), which may be important for explaining how relatively weak orbital forcing is amplified
by the climate system. The noise level varied on glacial-interglacial time scales and was enhanced
during glacial stages and suppressed during interglacials (Hodell et al., 2015), exhibiting characteristics
of an excitable system. The interaction of millennial- and orbital-scale climate variability may be an
important missing element for explaining the observed patterns of Quaternary climate change.

### 5. Conclusions

Site U1308 provides a 3.2-Myr record documenting the increase in the intensity of Northern

Hemisphere glaciation with mode changes at ~2.7, 1.5, 0.9 and 0.65 Ma. The 2.7-Ma transition (MIS



G6) marked the appearance of IRD at Site U1308 when benthic $\delta^{18}$O first exceeded 3.5 ‰. The event

was also associated with a strong decrease in benthic $\delta^{13}$C signalling a shoaling of the overturning

circulation cell and increased influence of southern sourced waters in the deep North Atlantic. Eurasian

ice sheets may have had a disproportionally large impact on deep-water circulation during the early

Pleistocene because of their proximity to deep-water source areas in the Nordic Seas. The carbon

isotope gradient between intermediate and deep-water increased at ~2.75 Ma (MIS G6) marking the

development a chemical divide in the Atlantic between well-ventilated intermediate water and more

poorly-ventilated deep water (Hodell and Venz-Curtis, 2006). Increased deep-sea carbon storage

resulted in $CO_2$ decline (Martinez-Boti et al., 2015) that, together with an exceptional eccentricity

minimum, ushered in the glacial-interglacial cycles of the Quaternary.

At Site U1308, the variability in the suborbital (millennial–scale) band began to increase at 1.8 Ma and

then strengthened significantly after ~1.5 Ma. From 1.5 Ma onward, millennial-scale IRD events

became a persistent feature of the Site U1308 record during glacial periods (Fig. 5). The 1.5-Ma

transition was also associated with reduction of NCW in the deep North Atlantic during glacials

(Raymo et al., 1990), and an increase in vertical carbon isotope gradients between the intermediate and

deep ocean, suggesting changes in deep carbon storage (Hodell and Venz-Curtis, 2006). Glacial-

interglacial climate changes became synchronized between the subpolar North and South Atlantic at

1.5 Ma (Hodell and Venz, 1992), and $CO_2$ variations may have assumed an increasingly important role

in the synchronization of glacial-interglacial climate change.

No major change is observed at Site U1308 when global ice volume increased at 0.9 Ma; however, at

Site 982 (57.5°N) MIS22-23-24 is marked by the first lows in bulk carbonate $\delta^{18}$O indicating delivery

of reworked carbonate (Fig. 10), which was likely sourced from the Eurasian ice sheet. We suggest

that the 0.9-Ma event may not have involved a major change in the stability of the Laurentide ice sheet

as there is no evidence for a Heinrich event associated with MIS 22, but rather the ice volume increase

may have been concentrated on Eurasia and Antarctica (Elderfield et al., 2012). The onset of glaciation

in the circum-Baltic region and in Alpine Europe is associated with MIS 22 and 20 (Head and Gibbard,

2015).



The time of greatest change in physical properties in the Site U1308 record occurred at 0.65 Ma (MIS

16) when Heinrich events first appeared chronicling a fundamental change in the dynamics of the

Laurentide Ice Sheet (Hodell et al., 2008). The intensity of IRD events increased but the frequency

decreased at this time, perhaps related to the formation of ice shelves in the Labrador Sea that acted as

an IRD filter (Alley et al., 2005). The growth of very large ice sheets on North America and the

appearance of Heinrich events in North Atlantic sediments coincided with the emergence of the saw-

tooth pattern of $\delta^{18}O$ change and strong quasi-periodic 100 kyr cycles.

We infer that orbital and millennial variability co-evolved during the Quaternary such that millennial

variability generally increased in intensity as ice sheets grew larger in size. Millennial variability

provides a source of short-term variability ("agitation") to the climate system that may play an

important role in glacial-interglacial climate transitions.   The strong link between IRD events and

decreases in benthic $\delta^{13}C$ supports a connection between ice sheet variability and deep ocean

circulation.  Furthermore, the $\Delta^{13}C$ gradient between intermediate and deep-water increased at the 2.7

and 1.5-Ma transitions (Hodell and Venz-Curtis, 2006), which may reflect an increasingly important

role of deep-sea carbon storage and atmospheric $CO_2$ in glacial-interglacial cycles through the

Quaternary.

*Author contributions.* Both authors were involved in the planning and execution of IODP Expedition

303 during which IODP Site U1308 was recovered., The authors have contributed equally to data

collection, interpretation and writing of the manuscript.

*Acknowledgments.* We thank Paul Wilson for providing splits of samples below 185 mcd for bulk

carbonate analysis and Simon Crowhurst for help with the wavelet analysis.  Jason Curtis, James Rolfe

and John Nicolson are thanked for analytical assistance in measuring stable isotopes. We gratefully

acknowledge assistance from the scientific party and crew of R/V JOIDES Resolution during IODP

Expedition 303, and from the curatorial staff at the IODP core repository in Bremen. Research funded

by NSF grants 0850413 and 1014506 to Channell, and NERC NE/H009930/1 to Hodell.






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



**Figure Captions**

Figure 1. Location of sites discussed in text (red circles) relative to reconstructed ice sheet extent in the Northern Hemisphere (continuous solid black line) showing the inferred locations of ice streams (black arrows; after Denton and Hughes, 1981) at the Last Glacial Maximum. Yellow stars indicate modern regions of deep water formation and green star marks the approximate location intermediate water formation during glacial times. This map emphasizes the point that most of the water stored in the North American and European ice sheets was discharged into the Atlantic Ocean in proximity to areas of deep water formation. Figure modified after Stokes and Clark (2001).

Figure 2. Oxygen isotope and magetostratigraphy of IODP Site U1308. Benthic $\delta^{18}O$ (black) was correlated to the LR04 stack (orange; Lisiecki and Raymo, 2005) and selected glacial marine isotope stages (MIS) are labeled. The oxygen isotope stratigraphy is complete except for a hiatus that has removed MIS G1 and G2 (~2.6 to 2.65 Ma). Inclination of natural remanent magetization (NRM) with polarity chrons, subchrons and excursions labelled (Channell et al., 2016).

Figure 3. The benthic oxygen and carbon isotope records for IODP Site U1308 (black) and DSDP Site 607 (gold; Ruddiman et al., 1986; Raymo et al., 1986). Prominent marine isotope stages are numbered and $\delta^{18}O$ values corresponding to MIS 5b, 4 and 2 are marked for reference. The orange arrows on the benthic $\delta^{13}C$ record highlights two trends of decreasing glacial $\delta^{13}C$ from MIS 22 to 14 and again from MIS 12 to 2 originally described by Raymo et al. (1990).

Figure 4. $\delta^{18}O$ of bulk carbonate (lower) and the difference between benthic and bulk $\delta^{18}O$ (upper) shown relative to the benthic $\delta^{18}O$ record (middle) of Site U1308. Prominent marine isotope stages are numbered. Top and bottom panels are continuous wavelet transform of the density and bulk $\delta^{18}O$ records of Site U1308, respectively. Black contour on the wavelet plots indicates the 5% significance level against red noise. The horizontal dashed white lines represent the orbital periods of precession (21 kyr), obliquity (41 kyr), and eccentricity (100 kyr).

Figure 5. Multitaper method power spectrum of bulk $\delta^{18}O$ using REDFIT [Schulz and Mudelsee, 2002]. The signal contains particularly strong power at 41, 19-21, 12.8, 9.7 and 9 kyrs. The power at 21 and 19 kyrs may not entirely represent precession because in the 41-kyr world, bulk $\delta^{18}O$ minima tend to occur at the onset and end of glacial cycles, which can result in a semi-obliquity peak near 20 kyrs.

Figure 6. Natural gamma radiation (NGR; green) and density (gray) relative to the benthic $\delta^{18}O$ record of Site U1308. Prominent marine isotope stages are numbered. Top and bottom panels are continuous wavelet transform of the NGR and density records of Site U1308, respectively. Black contour on the





wavelet plots indicates the 5% significance level against red noise. The horizontal dashed white lines

represent the orbital periods of precession (21 kyr), obliquity (41 kyr), and eccentricity (100 kyr).

Figure 7. $\kappa_{ARM}/\kappa$ (proxy for magnetic grain size) and magnetic susceptibility (brown) relative to the

benthic $\delta^{18}O$ record of Site U1308. Prominent marine isotope stages are numbered. Top and bottom

panels are continuous wavelet transform of the $\kappa_{ARM}/\kappa$ and magnetic susceptibility records of Site

U1308, respectively. Black contour on the wavelet plots indicates the 5% significance level against red

noise. The horizontal dashed white lines represent the orbital periods of precession (21 kyr), obliquity

(41 kyr), and eccentricity (100 kyr).


Figure 8. Alkenone ($U^{k'}_{37}$) sea surface temperature at Site U1313 (gold, Naafs et al., 2012) and 982

(blue; Lawrence et al., 2009). The SST records were interpolated at 1 kyr sampling interval and the

U1313 record was subtracted for 982.

Figure 9. Bulk carbonate $\delta^{18}O$ at Site U1308 (black) compared to the Quartz/Calcite (Qtz/Cal, blue),

Dolomite/Calcite (Dol/Cal, red) and occurrence of C28 steroids (a petrogenic compound indicative of

input from Hudson Strait) at Site U1313 (Naafs et al., 2013).

Figure 10. Comparison of the deconvolved oxygen isotope signal of seawater (d18Owater) from Site

1123 (red) (Elderfield et al., 2012) and the $\delta^{18}O$ record of the shallow-dwelling *Globigerinoides*

*sacculifer* from Site 806 on the Ontong Java Plateau (gray) (Berger et al., 1993, 1994). Note the abrupt

increase in $\delta^{18}O$ of glacial stages following 900ka after MIS 22.

Figure 11. Bulk carbonate $\delta^{18}O$ (lower) and magnetic susceptibility (upper) at Site 982 compared to the

LR04 benthic $\delta^{18}O$ record (middle; Lisiecki and Raymo, 2005). Note the large millennial decreases in

bulk carbonate $\delta^{18}O$ beginning in MIS 22-24 at 900 ka.

Figure 12. Natural Gamma Radiation (NGR) and benthic $\delta^{18}O$ at Site U1304 on the southernmost

Gardar Drift. Variability of the NGR signal suggests more frequent millennial-scale variability prior to

650 ka (MIS 16) when the climate system spends more time in an intermediate ice volume state.

Magnetostratigraphy is after Xuan et al. (submitted). The age model for the past 1 Ma is based on

correlation of the benthic $\delta^{18}O$ record to LR04 and assuming constant sedimentation rates between the

ages of polarity reversals thereafter.

1155

Figure 13. (A) The difference between benthic and bulk carbonate $\delta^{18}O$ at Site U1308. (B) Iron (red)

and dust (black) accumulation at Site 1090 in the South Atlantic (Martinez-Garcia et al., 2011) (C) The



carbon isotope gradient between intermediate and deep-water ($\Delta^{13}C_{ID}$) in the Southern Ocean (Hodell

1160   and Venz, 2006). (D) The LR04 benthic $\delta^{18}O$ record (Lisiecki and Raymo, 2005).



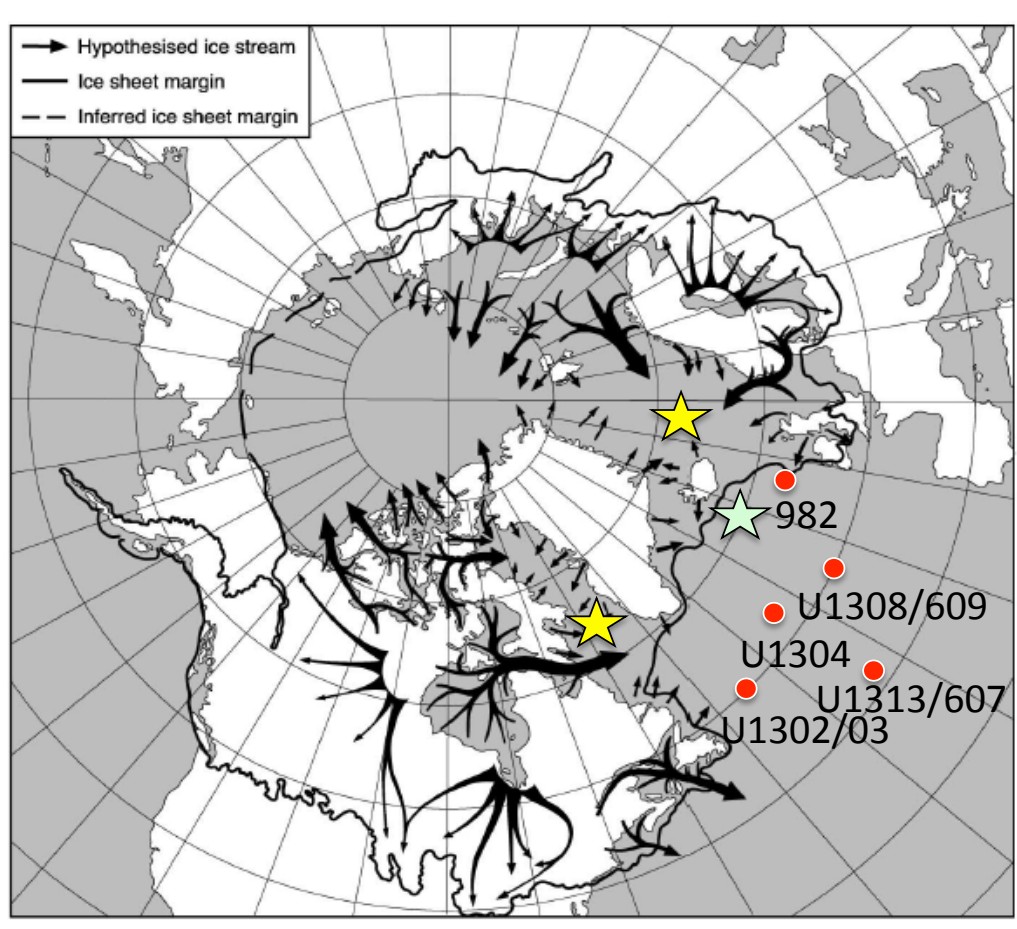

fig01



fig02







fig03

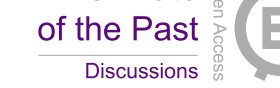



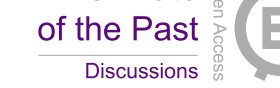

fig04





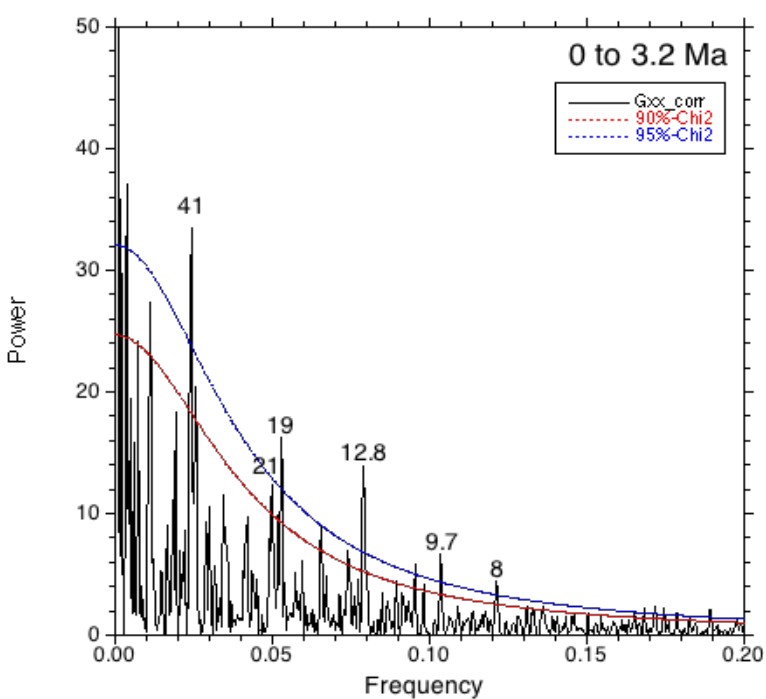

fig05



Age (ka)

fig06




fig07



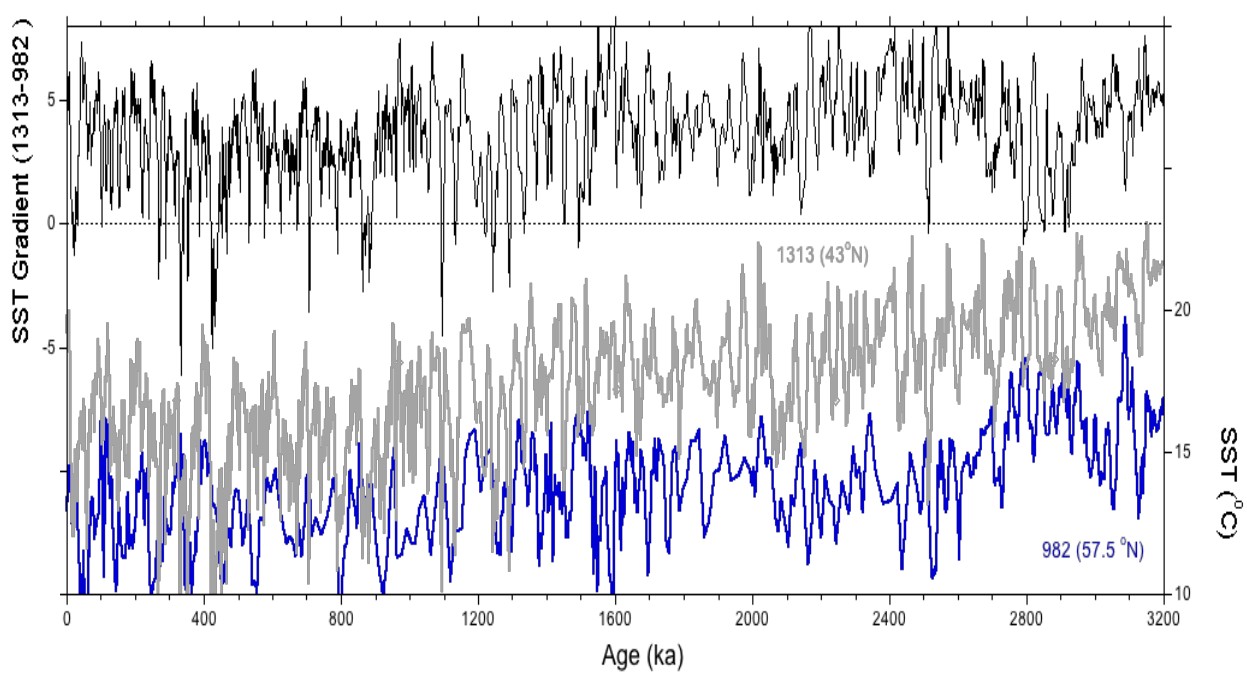

fig08





fig09





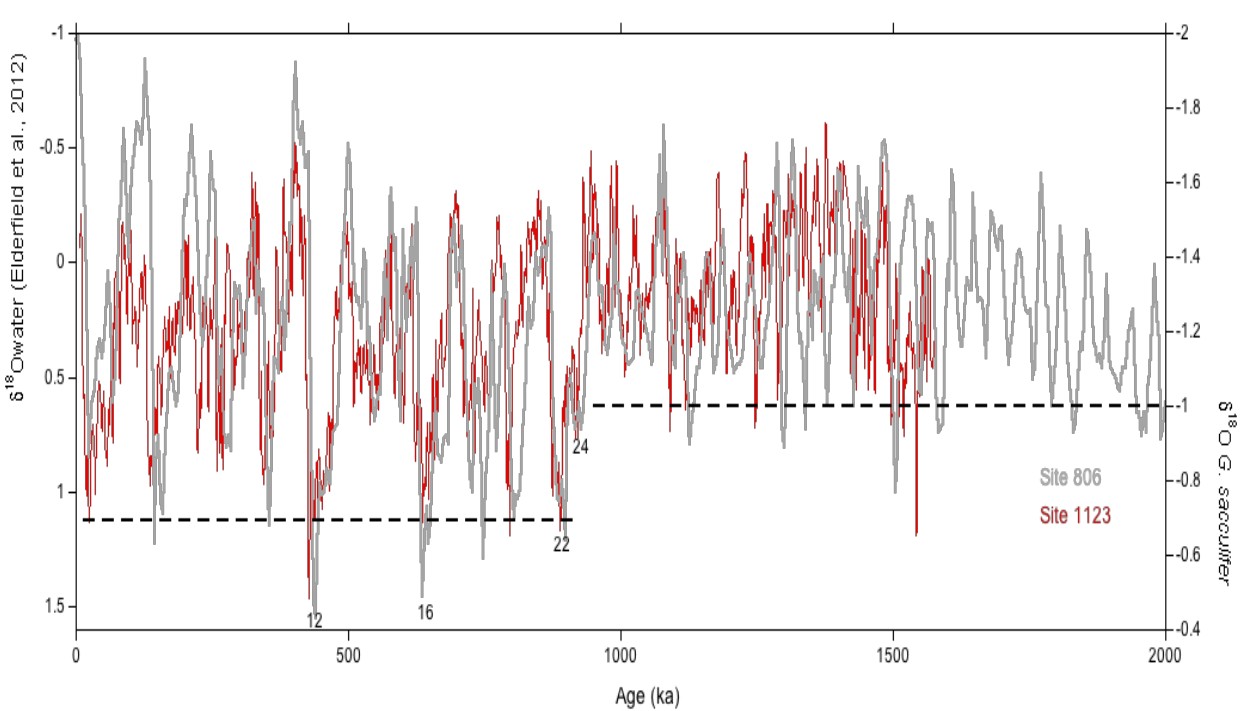

fig10





fig11



fig12





fig13





Table 1. Age-depth control points used to establish chronology at Site U1308.

| Depth [mcd] | Age model [ka] | Age model type | Reference |
|---|---|---|---|
| 0.953 | 17.79 | radiocarbon | Obrochta et al. (2012, 2014) |
| 0.981 | 19.17 | radiocarbon | Obrochta et al. (2012, 2014) |
| 1.007 | 19.56 | radiocarbon | Obrochta et al. (2012, 2014) |
| 1.076 | 20.18 | radiocarbon | Obrochta et al. (2012, 2014) |
| 1.167 | 22.66 | radiocarbon | Obrochta et al. (2012, 2014) |
| 1.220 | 23.90 | radiocarbon | Obrochta et al. (2012, 2014) |
| 1.238 | 24.58 | radiocarbon | Obrochta et al. (2012, 2014) |
| 1.256 | 25.28 | radiocarbon | Obrochta et al. (2012, 2014) |
| 1.305 | 25.59 | radiocarbon | Obrochta et al. (2012, 2014) |
| 1.335 | 27.05 | radiocarbon | Obrochta et al. (2012, 2014) |
| 1.555 | 30.10 | radiocarbon | Obrochta et al. (2012, 2014) |
| 1.635 | 30.89 | radiocarbon | Obrochta et al. (2012, 2014) |
| 1.700 | 31.98 | radiocarbon | Obrochta et al. (2012, 2014) |
| 1.886 | 33.20 | radiocarbon | Obrochta et al. (2012, 2014) |
| 1.988 | 33.92 | radiocarbon | Obrochta et al. (2012, 2014) |
| 2.251 | 35.45 | %Nps | Obrochta et al. (2012, 2014) |
| 2.512 | 38.15 | %Nps | Obrochta et al. (2012, 2014) |
| 2.917 | 41.45 | %Nps | Obrochta et al. (2012, 2014) |
| 3.060 | 43.30 | %Nps | Obrochta et al. (2012, 2014) |
| 3.324 | 46.65 | %Nps | Obrochta et al. (2012, 2014) |
| 4.099 | 54.40 | %Nps | Obrochta et al. (2012, 2014) |
| 4.308 | 57.80 | %Nps | Obrochta et al. (2012, 2014) |
| 4.360 | 59.05 | %Nps | Obrochta et al. (2012, 2014) |
| 4.526 | 63.95 | %Nps | Obrochta et al. (2012, 2014) |
| 4.734 | 69.60 | %Nps | Obrochta et al. (2012, 2014) |
| 4.885 | 72.25 | %Nps | Obrochta et al. (2012, 2014) |
| 5.163 | 76.50 | %Nps | Obrochta et al. (2012, 2014) |
| 5.900 | 86.00 | LR04 | Hodell et al. (2008) |
| 6.220 | 93.00 | LR04 | Hodell et al. (2008) |
| 7.060 | 104.00 | LR04 | Hodell et al. (2008) |
| 8.750 | 127.00 | LR04 | Hodell et al. (2008) |
| 9.170 | 136.00 | LR04 | Hodell et al. (2008) |
| 11.540 | 191.00 | LR04 | Hodell et al. (2008) |
| 12.420 | 201.00 | LR04 | Hodell et al. (2008) |
| 13.650 | 220.00 | LR04 | Hodell et al. (2008) |
| 15.320 | 243.00 | LR04 | Hodell et al. (2008) |
| 18.060 | 281.00 | LR04 | Hodell et al. (2008) |
| 18.740 | 292.00 | LR04 | Hodell et al. (2008) |
| 21.420 | 329.40 | LR04 | this study |
| 21.850 | 341.80 | LR04 | this study |
| 24.730 | 392.00 | LR04 | Hodell et al. (2008) |
| 27.400 | 431.00 | LR04 | Hodell et al. (2008) |





| | | | |
|---|---|---|---|
| 29.060 | 475.00 | LR04 | Hodell et al. (2008) |
| 32.680 | 513.00 | LR04 | Hodell et al. (2008) |
| 34.920 | 534.00 | LR04 | Hodell et al. (2008) |
| 37.560 | 580.00 | LR04 | Hodell et al. (2008) |
| 38.580 | 596.00 | LR04 | Hodell et al. (2008) |
| 39.880 | 622.00 | LR04 | Hodell et al. (2008) |
| 43.720 | 690.00 | LR04 | Hodell et al. (2008) |
| 47.240 | 726.00 | LR04 | Hodell et al. (2008) |
| 48.720 | 746.00 | LR04 | Hodell et al. (2008) |
| 51.140 | 788.00 | LR04 | Hodell et al. (2008) |
| 52.580 | 814.00 | LR04 | Hodell et al. (2008) |
| 56.410 | 866.00 | LR04 | Hodell et al. (2008) |
| 58.520 | 902.00 | LR04 | Hodell et al. (2008) |
| 60.200 | 922.00 | LR04 | Hodell et al. (2008) |
| 64.370 | 964.00 | LR04 | Hodell et al. (2008) |
| 65.710 | 978.00 | LR04 | Hodell et al. (2008) |
| 65.970 | 984.00 | LR04 | Hodell et al. (2008) |
| 67.730 | 998.00 | LR04 | Hodell et al. (2008) |
| 68.050 | 1004.00 | LR04 | Hodell et al. (2008) |
| 70.410 | 1032.00 | LR04 | Hodell et al. (2008) |
| 72.070 | 1058.00 | LR04 | Hodell et al. (2008) |
| 72.770 | 1070.00 | LR04 | Hodell et al. (2008) |
| 77.250 | 1098.00 | LR04 | Hodell et al. (2008) |
| 78.250 | 1108.00 | LR04 | Hodell et al. (2008) |
| 79.990 | 1126.00 | LR04 | Hodell et al. (2008) |
| 80.750 | 1132.00 | LR04 | Hodell et al. (2008) |
| 86.100 | 1200.00 | LR04 | Hodell et al. (2008) |
| 87.200 | 1218.00 | LR04 | Hodell et al. (2008) |
| 87.560 | 1222.00 | LR04 | Hodell et al. (2008) |
| 89.900 | 1250.00 | LR04 | Hodell et al. (2008) |
| 91.860 | 1290.00 | LR04 | Hodell et al. (2008) |
| 94.380 | 1316.00 | LR04 | Hodell et al. (2008) |
| 96.620 | 1340.00 | LR04 | Hodell et al. (2008) |
| 98.200 | 1354.00 | LR04 | Hodell et al. (2008) |
| 100.680 | 1372.00 | LR04 | Hodell et al. (2008) |
| 102.200 | 1398.00 | LR04 | Hodell et al. (2008) |
| 102.960 | 1406.00 | LR04 | Hodell et al. (2008) |
| 103.710 | 1412.00 | LR04 | Hodell et al. (2008) |
| 103.960 | 1418.00 | LR04 | Hodell et al. (2008) |
| 104.670 | 1426.00 | LR04 | Hodell et al. (2008) |
| 105.630 | 1442.00 | LR04 | Hodell et al. (2008) |
| 106.310 | 1448.00 | LR04 | Hodell et al. (2008) |
| 106.460 | 1456.00 | LR04 | Hodell et al. (2008) |
| 108.710 | 1488.00 | LR04 | Hodell et al. (2008) |
| 109.040 | 1496.00 | LR04 | Hodell et al. (2008) |





| | | | |
|---|---|---|---|
| 109.990 | 1510.00 | LR04 | Hodell et al. (2008) |
| 112.500 | 1530.50 | LR04 | Channell et al. (2016) |
| 113.750 | 1546.90 | LR04 | Channell et al. (2016) |
| 115.420 | 1571.10 | LR04 | Channell et al. (2016) |
| 116.940 | 1584.40 | LR04 | Channell et al. (2016) |
| 119.030 | 1610.90 | LR04 | Channell et al. (2016) |
| 119.750 | 1618.80 | LR04 | Channell et al. (2016) |
| 120.250 | 1625.00 | LR04 | Channell et al. (2016) |
| 120.750 | 1628.90 | LR04 | Channell et al. (2016) |
| 122.000 | 1641.40 | LR04 | Channell et al. (2016) |
| 123.920 | 1666.40 | LR04 | Channell et al. (2016) |
| 128.500 | 1698.40 | LR04 | Channell et al. (2016) |
| 129.160 | 1710.80 | LR04 | Channell et al. (2016) |
| 130.750 | 1743.30 | LR04 | Channell et al. (2016) |
| 131.480 | 1752.50 | LR04 | Channell et al. (2016) |
| 135.330 | 1787.50 | LR04 | Channell et al. (2016) |
| 136.250 | 1800.80 | LR04 | Channell et al. (2016) |
| 138.020 | 1815.80 | LR04 | Channell et al. (2016) |
| 139.000 | 1826.70 | LR04 | Channell et al. (2016) |
| 140.340 | 1859.20 | LR04 | Channell et al. (2016) |
| 140.650 | 1870.80 | LR04 | Channell et al. (2016) |
| 145.170 | 1898.30 | LR04 | Channell et al. (2016) |
| 146.030 | 1906.70 | LR04 | Channell et al. (2016) |
| 146.760 | 1913.30 | LR04 | Channell et al. (2016) |
| 148.780 | 1941.70 | LR04 | Channell et al. (2016) |
| 150.260 | 1964.50 | LR04 | Channell et al. (2016) |
| 151.970 | 1998.00 | LR04 | Channell et al. (2016) |
| 153.070 | 2009.00 | LR04 | Channell et al. (2016) |
| 157.040 | 2044.00 | LR04 | Channell et al. (2016) |
| 157.650 | 2047.60 | LR04 | Channell et al. (2016) |
| 159.550 | 2062.90 | LR04 | Channell et al. (2016) |
| 160.770 | 2086.30 | LR04 | Channell et al. (2016) |
| 162.120 | 2116.10 | LR04 | Channell et al. (2016) |
| 162.540 | 2123.40 | LR04 | Channell et al. (2016) |
| 164.190 | 2145.30 | LR04 | Channell et al. (2016) |
| 165.290 | 2167.20 | LR04 | Channell et al. (2016) |
| 167.130 | 2192.70 | LR04 | Channell et al. (2016) |
| 168.190 | 2205.30 | LR04 | Channell et al. (2016) |
| 169.790 | 2236.60 | LR04 | Channell et al. (2016) |
| 170.830 | 2250.30 | LR04 | Channell et al. (2016) |
| 174.200 | 2273.70 | LR04 | Channell et al. (2016) |
| 175.210 | 2290.00 | LR04 | Channell et al. (2016) |
| 176.600 | 2309.50 | LR04 | Channell et al. (2016) |
| 178.680 | 2332.90 | LR04 | Channell et al. (2016) |
| 180.140 | 2349.90 | LR04 | Channell et al. (2016) |





| | | | |
|---|---|---|---|
| 181.600 | 2373.30 | LR04 | Channell et al. (2016) |
| 184.310 | 2388.40 | LR04 | Channell et al. (2016) |
| 186.000 | 2406.00 | LR04 | Channell et al. (2016) |
| 188.950 | 2428.80 | LR04 | Channell et al. (2016) |
| 189.150 | 2434.60 | LR04 | Channell et al. (2016) |
| 189.350 | 2463.30 | LR04 | Channell et al. (2016) |
| 189.900 | 2487.40 | LR04 | Channell et al. (2016) |
| 190.700 | 2500.40 | LR04 | Channell et al. (2016) |
| 190.950 | 2520.60 | LR04 | Channell et al. (2016) |
| 191.550 | 2540.10 | LR04 | Channell et al. (2016) |
| 193.150 | 2553.80 | LR04 | Channell et al. (2016) |
| 194.800 | 2571.40 | LR04 | Channell et al. (2016) |
| 196.850 | 2594.80 | LR04 | Channell et al. (2016) |
| 197.400 | 2601.90 | LR04 | Channell et al. (2016) |
| 197.710 | 2650.60 | LR04 | Channell et al. (2016) |
| 199.580 | 2681.20 | LR04 | Channell et al. (2016) |
| 200.490 | 2689.40 | LR04 | Channell et al. (2016) |
| 201.860 | 2704.40 | LR04 | Channell et al. (2016) |
| 203.540 | 2730.60 | LR04 | Channell et al. (2016) |
| 210.690 | 2799.40 | LR04 | Channell et al. (2016) |
| 210.900 | 2805.00 | LR04 | Channell et al. (2016) |
| 211.420 | 2819.50 | LR04 | Channell et al. (2016) |
| 213.080 | 2840.60 | LR04 | Channell et al. (2016) |
| 213.660 | 2847.70 | LR04 | Channell et al. (2016) |
| 213.750 | 2857.00 | LR04 | Channell et al. (2016) |
| 216.420 | 2864.80 | LR04 | Channell et al. (2016) |
| 220.250 | 2876.60 | LR04 | Channell et al. (2016) |
| 221.600 | 2893.00 | LR04 | Channell et al. (2016) |
| 224.250 | 2913.30 | LR04 | Channell et al. (2016) |
| 225.670 | 2936.70 | LR04 | Channell et al. (2016) |
| 227.580 | 2957.00 | LR04 | Channell et al. (2016) |
| 229.080 | 2967.20 | LR04 | Channell et al. (2016) |
| 231.420 | 2982.00 | LR04 | Channell et al. (2016) |
| 232.580 | 2999.20 | LR04 | Channell et al. (2016) |
| 234.000 | 3015.60 | LR04 | Channell et al. (2016) |
| 235.080 | 3025.80 | LR04 | Channell et al. (2016) |
| 237.000 | 3040.60 | LR04 | Channell et al. (2016) |
| 237.500 | 3048.40 | LR04 | Channell et al. (2016) |
| 238.890 | 3059.90 | LR04 | Channell et al. (2016) |
| 240.500 | 3084.90 | LR04 | Channell et al. (2016) |
| 241.720 | 3092.70 | LR04 | Channell et al. (2016) |
| 247.330 | 3134.90 | LR04 | Channell et al. (2016) |
| 254.920 | 3207.00 | LR04 | Channell et al. (2016) |
| 263.170 | 3330.00 | LR04 | Channell et al. (2016) |