# Peer review of "Mode transitions in Northern Hemisphere Glaciation: Co-evolution of millennial and orbital variability in Quaternary climate"

_Climate of the Past, 2016_

## Referee Comment (RC1) · I. Bailey (Referee) · 5 Apr 2016

Hodell & Channell present substantial new stable isotope datasets measured on benthic foraminifera and on bulk carbonate from North Atlantic IODP Site U1308. They use these data, together with physical property measurements and previously published U1308 magnetics data, to shed new light on the evolution of Quaternary North Atlantic climate on orbital to suborbital timescales. They propose that orbital- and millennial-scale variability centred on the North Atlantic 'co-evolved' during the Quaternary and link this evolution to a series of mode transitions in climate at ∼2.7, ∼1.5,

~0.9 and ~0.65 Ma. In presenting this work, the authors seemingly bring to fruition one important goal of Expedition 303/306: to document the evolution of Quaternary millennial-scale climate variability recorded at U1308, the reoccupation of DSDP Site 609. Studies of 609 set the agenda for our understanding of abrupt and rapid North Atlantic climate change during the last glacial and it is fitting that its reoccupation is proving to be just as important for advancing our understanding of these issues. It is certainly nice to review a paper in which I cannot find any real problems. For me its publication in CP is a formality following minor revision. The attached pdf contains a series of minor comments that I would like to see addressed to help improve their contribution further still.

Best Wishes, Ian Bailey

Please also note the supplement to this comment:
http://www.clim-past-discuss.net/cp-2016-30/cp-2016-30-RC1-supplement.pdf

———————————————————

[Figure]

**Supplement:**

**Review of Hodell and Channell CPD**

Hodell & Channell present substantial new stable isotope datasets measured on benthic foraminifera and on bulk carbonate from North Atlantic IODP Site U1308. They use these data, together with physical property measurements and previously published U1308 magnetics data, to shed new light on the evolution of Quaternary North Atlantic climate on orbital to suborbital timescales. They propose that orbital- and millennial-scale variability centred on the North Atlantic 'co-evolved' during the Quaternary and link this evolution to a series of mode transitions in climate at ~2.7, ~1.5, ~0.9 and ~0.65 Ma. In presenting this work, the authors seemingly bring to fruition one important goal of Expedition 303/306: to document the evolution of Quaternary millennial-scale climate variability recorded at U1308, the reoccupation of DSDP Site 609. Studies of 609 set the agenda for our understanding of abrupt and rapid North Atlantic climate change during the last glacial and it is fitting that its reoccupation is proving to be just as important for advancing our understanding of these issues. It is certainly nice to review a paper in which I cannot find any real problems. For me its publication in CP is a formality following minor revision. The attached pdf contains a series of minor comments that I would like to see addressed to help improve their contribution further still.

Best Wishes, Ian Bailey

**Line by line comments**

Line 35: For those less familiar with DSDP/(I)ODP best to spell these acronyms out in full here.

Line 37: might help to add a time in Ma in parentheses after 'latest Pliocene'.

Line 113: Was this modification made by Channell et al. (2016) or was it this study? There seems to be a consistent 9 cm depth offset between the depths assigned to the ages (so for depths greater than ~100 m) presented in Tables 1 of this Ms and of Channell et al. (2016). I apologise if I've got this wrong, but if I've read the tables correctly, does this represent an a slight modification of the Channell et al. (2016) age model in this Ms?

Section 2.3: Do all the benthic $\partial^{18}$O data come from Hodell et al. (2008) and Channell et al. (2016)? Is what you present here for the first time the associated $\partial^{13}$C data for the older than ~1.5 Ma interval? If so, you could save text by simply saying you utilise previously published stratigraphies based on benthic $\partial^{18}$O and present a new benthic $\partial^{13}$C record from the >1.5 Ma samples analysed by Channell et al. (2016) that extends the previously published $\partial^{13}$C from Hodell et al. (2008) back to ~3 Ma.

Your comparison of 607-U1308 stable isotope data in Section 3.1 Ma would benefit from using $\partial^{18}$O/$\partial^{13}$C splices for 607/U1313 (so using the Bolton et al. (2010)/Lang et al. (2014) data for >2.4 Ma). The U1313 stable isotope records for ~3.3-2.4 Ma are twice the resolution of the 607 record for this time, and using these data will modify some statements you make in this section. Both datasets can be found on Pangaea, but please feel free to email me and I can provide you with a copy.

Lines 189–190: Is U1308 $\partial^{13}$C really typically that much more negative than that from U1313 during MIS G6? It's hard to see the detail in your Figure 3, but it looks as though the much more U1308 negative signal can be attributed to two data points. Instead it seems that most of the time U1308 $\partial^{13}$C is only ~0.2-3‰ lighter than at U1313 during G6. This difference may point towards some fundamental difference in source/aging of $\partial^{13}$C at the deeper U1308 relative to 607. During G6 the $\partial^{13}$C gradient between U1313 and records of end-member NCW $\partial^{13}$C (e.g. potentially assessed from Site 982) is still relatively large (Lang et al., 2016). If there is significant SCW at U1313 during MIS G6 in the deep (3.4 km) western North Atlantic, then the lower $\partial^{13}$C values at the deeper (3.8 km), albeit more northerly eastern basin Site U1308 may reflect that there is a stronger SCW influence at U1308 than at U1313 during MIS G6 (is that really likely?). Alternatively, the waters bathing U1308

may be dense overflow waters from the north (Bell et al., 2015), although the similar $\partial^{18}O$ values at U1313 and U1308 during this glacial would suggest otherwise.

Line 223–225: Do LGM iceberg drift models of Grant Biggs and Ros D'Eath support the notion that British Chalk/Scandinavian rocks might be a notable source of IRD to U1308? Don't they show it is unlikely that many Scandinavian icebergs/IRD would reach south of Iceland.

Line 235: $\partial^{18}O$ (benthic – bulk) increase during MIS 82 is consistent with the fact that this glacial may be characterised by the first late Pleistocene-magnitude sea-level fall Rohling et al. (2014).

Line 245: the sentence here reads as though you are saying that there is Ca/Sr data in Figure 4.

Line 323: Bailey et al. (2012) is a good reference for North Atlantic IRD sources during MIS 100, but the key reference for evidence of a dominantly Archaean provenance for North Atlantic IRD prior to MIS 100 should be Bailey et al. (2013) where that observation was published for the first time.

Line 330: Raymo et al. (1992) interpret a divergence in Site 607 $\partial^{13}C$ towards values more negative than that of Site 552 during MIS 100 as the first evidence for decreased NCW in the deep North Atlantic Ocean during iNHG. That view has been updated recently in Lang et al. (2016), since it seems that based on U1313 $\partial^{13}C$ and fish debris $\varepsilon_{Nd}$ that MIS G6 is the first glacial associated with significant (and potentially LGM magnitude) SCW incursion into the deep North Atlantic Ocean.

Line 341: Perhaps cite Rohling et al. (2014) here for magnitude of MIS 82 glaciation.

Lines 342–344: We didn't find sand IRD in U1308 sediments for MIS G6 when studying at a 30 cm sampling resolution (Bailey et al., 2010). Bolton et al. (2010) and Lang et al. (2014) have shown through higher resolution analyses (every 5 cm) that sand IRD is similarly absent in sediments deposited at U1313 until MIS G4, but that values of ~40 grains gram (comparable to the LG scenario at U1313 outside of H-events; Lang et al., 2016) do not occur at this site until MIS 100. These more recent studies have updated the view of when significant icebergs arrived at 40°N based on DSDP studies (Raymo et al., 1986; Kleiven et al., 2002) and support what your data show, i.e. that widespread iceberg rafting and IRD deposition across the North Atlantic Ocean did not occur until MIS 100…reflecting the true large magnitude of that NH glaciation relative to previous cold stages (as potentially also confirmed by, e.g. Balco and Rovey, 2010; Brigham-Grette et al., 2013).

Line 345: MIS 94-52 broadly coincides with inference on increased AMOC strength by Bell et al. (2015). Do your eastern basin U1308 $\partial^{13}C$ data support the Bell et al. interpretation, or suggest an alternative origin of the Walvis Ridge 'overflow' signal they report?

Line 352: the low benthic $\partial^{13}C$ values you report for U1308 in the early Pleistocene should be discussed in the context of the ideas of Bell et al. (2015). You may end up dismissing this suggestion (if you haven't already), but I think this is worth considering because Site 607 doesn't record significant evidence for major shoaling of NCW between 1.5-2 Ma (Lang et al., 2016), and you think it would do if FIS meltwater was impacting significantly on NADW production at this time. The NCW cell can shoal and AMOC can remain relatively strong, but models suggest that if AMOC is reduced then the NCW cell has to shoal. Can we rule out productivity aging of benthic $\partial^{13}C$ at U1308?

Line 368: A obvious question here is "based on the records we've got so far, does it look as though the magnitude and spatial fingerprint of suborbital climate change observed for MIS 3 replicated at any other time during the past ~3 Ma?" The short answer is probably yes, with evidence for DO events as far south as 30°N since ~0.9 Ma (Ferretti et al., 2010; Weirauch et al., 2008). Prior to this time strong evidence exists for DO-like events during MIS 40 and 38 (~1.3 Ma) at 37°N in the

northeastern equatorial Atlantic (Birner et al., 2016), but seemingly not at 30˚N in the northwestern North Atlantic (Weirauch et al., 2008). No convincing evidence exists anywhere yet for DO–magnitude change during any earliest Pleistocene glacials, e.g. during MIS 100 (one of the more well studied cold stages for this time), but instead muted suborbital change in planktic $\partial^{18}O$ and SST ~40-60˚N (Bartoli et al., 2006; Becker et al., 2006; Bolton et al., 2010; Friedrich et al., 2013).

Benthic $\partial^{18}O$ records suggest our planet's climate system has been crossing this +3.5‰ threshold during glacials ever since ~2.7-2.5 Ma. If we assume that this benthic $\partial^{18}O$ value corresponds to a relatively narrow range of NH ice-sheet growth then the available evidence suggests that the spatial fingerprint of DO-like change over the past 3 Ma is not consistent with the climate system responding in a repeatable (pseudo predictable) manner to it sitting in an intermediate ice-volume window. If it did, we should expect to see the same spatial pattern more or less emerging for amplification of suborbital climate change during all big benthic $\partial^{18}O$ glacials (>+3.5‰) from ~2.5 Ma. The occurrence of DO-like change is clearly linked to NH ice sheet size, but records of the 41-kyr world suggest to me it is too simplistic to think of it as a straight forward ice-volume feedback (or our understanding of NH ice sheet volume during the 41-kyr world needs revision). If as yet undiscovered DO-like magnitude change really is restricted to the highest latitudes during the earliest Pleistocene, then that suborbital change (and the mechanisms responsible for it) do not seem to be analogous to events during the LG.

Line 375–278: McIntye et al. (2001) present strong evidence for millennial-scale changes in iceberg rafting to Site 983 in the early Pleistocene (~1.93-1.75 Ma). You also know we found the same thing at U1313 and U1308 during the much older MIS 100 (Bolton et al., 2010; Bailey et al., 2010) and at U1308 during MIS G4 (Bailey et al., 2010), but none of these earliest Pleistocene events are yet found to be associated with large amplitude swings in SST/$\partial^{18}O$ (Becker et al., 2006; Bartoli et al., 2006; Bolton et al., 2010; Friedrich et al., 2013). The point I am trying to make here, and one you obviously appreciate, is that millennial-scale pulses of IRD deposition do not necessarily imply large magnitude swings in climate on such timescales, just that there are likely millennial-scale swings in climate driving the mass balance of ice-sheets/glacier at the coast at those times.

Marshall and Koutnik (2006) show that millennial-scale episodes of iceberg rafting can still be anticipated with muted suborbital climatic variability, but that such pulses might be set against a steadier background of IRD inputs, making them less distinct in the sediment record. If suborbital change during the earliest Pleistocene was muted relative to the late Pleistocene, we may therefore find that overall IRD inputs during earliest Pleistocene glacials were higher, but that suborbital-scale IRD pulses superimposed on this signal were muted, relative to IRD inputs during e.g. MIS 3 at U1308. Maybe it is best to look for this at a site further north where the iceberg/IRD survivability issue less strongly influences IRD inputs, but maybe worth thinking along these lines here since your record is the only suborbital proxy IRD record we have that spans the entire Quaternary.

Line 384/436/561: is Figure 5 the correct figure to cite here? Don't you mean Fig. 6 evol. power spec?

Line 391: please place a horizontal line at the benthic $\partial^{18}O$ value of +4 ‰ (~MIS 4) and +3.5 ‰ (McManus) to guide the reader's eye when they examine Figure 4.

Line 397: 'ice volume was about twice as great in North America compared to Eurasia'. I don't disagree that your datasets suggest that the deposition of HS-sourced material increased from 1.6 Ma (seems consistent with U1313 data from Naafs et al., 2013), but how do you then extend that to what seems like a relatively precise quantification of relative differences in ice volume?

Line 410: Good to plot an indicator of IRD in Figure 9 to help the reader see more easily the relationship between iceberg rafting to U1308 and U1313 and the SST gradient evolution.

Lines 412–426: again, see the recent findings of Lang et al. (2016) for new context on the pioneering observations of Raymo et al. (1990; 1992) and those made subsequently by e.g. Lisiecki (2014).

Lines 446–462: Have you compared your bulk $\partial^{18}$O record and/or $\partial^{18}$O (benthic – bulk) to Steve Barker's synthetic Greenland DO record? Is it may be worth showing a plot of this, if only in the supplementary guide. How does the variability in your record(s) for MIS 41-37 compare to those from your work in Birner et al. (2016)? Do we see evidence for the same number of ice-rafting events at U1308 as reported by Raymo et al. (1998) further north at ODP Site 983 during MIS 40 that you've tied convincingly to the DO-like variability seen in *G. bulloides* $\partial$18O from the Iberian Margin?

Lines 485–486: what's the Site 982 bulk $\partial^{18}$O data source? Are these data produced for this study? If so, please mention these analyses in your methods text. If not, the data source needs including.

**Figures**
Figure 1. Nice map. Perhaps state what the yellow/green triangles mean in your key too.

Figure 2. I think it would still help to have a key on the figure so it is easier for the reader to work out which record is the LR04 vs U1308 $\partial^{18}O_b$ (like you do in Figure 10).

Figure 3. A key showing which records are from 607 versus U1308 would aid the reader. I suggest labelling the horizontal lines with '21', '41' and '100' kyr. Ditto Figures 4, 6 and 7.

Figure 4. Please add the horizontal lines for MIS 5b, 4 and 2 onto the benthic $\partial^{18}$O data (as it is on Figure 3). Please also label the key bulk carbonate $\partial^{18}$O values referred to the in text, e.g. the -4 ‰ value characteristic of H-layers and the -2 ‰ value characteristic of DO-type ice-rafting events.

Figure 6. Given the density increase with depth, to make the suborbital events even clearer, it might be helpful to detrend the density data plotted in Fig. 6 by subtracting the linear best fit from it.

Maybe combine Figures 8 and 9 to help the reader see clearly how the 982-U1313 SST gradient evolves alongside changes in IRD inputs to these two sites.

Figure 13. Please label site names on dust records. Could also do with labelling key MIS on the LR04 or including vertical guide lines. It would also help to label all HS H-layers on the relevant figures to help tell apart HS-sourced H-layers and non-H-event (DO) IRD deposition in your bulk $\partial^{18}$O record.

Figure 12: Data sources for U1304 NGR and benthic $\partial^{18}$O not given in caption, or is it all presented in Xuan et al. (submitted)? If so a quick revision of the caption text is needed.

**Additional references cited:**
Bailey, I., Hole, G.M., Foster, G.L., Wilson, P.A., Storey, C.D., Trueman, C.N., Raymo, M.E. (2013). An alternative suggestion for the Pliocene onset of major northern hemisphere glaciation based on the geochemical provenance of North Atlantic Ocean ice-rafted debris. Quaternary Science Reviews 75, 181–194, http://dx.doi.org/10.1016/j.quascirev.2013.06.004.
Becker, J., Lourens, L.J., Raymo, M.E. (2006). High-frequency climate linkages between the North Atlantic and the Mediterranean during marine oxygen isotope stage 100 (MIS100). Paleoceanography 21, PA3002, doi:10.1029/2005PA001168.
Bell, D.B., Jung, S.J., Kroon, D. (2015). The Plio-Pleistocene development of Atlantic deep-water circulation and its influence on climate trends. Quaternary Science Reviews 123, 265-282. http://dx.doi.org/10.1016/j.quascirev.2015.06.026.
Brigham-Grette, J. et al. (2013). Pliocene Warmth, Polar Amplification, and Stepped Pleistocene Cooling Recorded in NE Arctic Russia.Science 340, 142. doi: 10.1126/science.1233137.
Ferretti, P., Crowhurst, S.J., Hall, M.A., Cacho, I. (2010). North Atlantic millennial-scale climate variability 910 to 790 ka and the role of the equatorial insolation forcing, Earth Planet. Sci. Lett. 293, 28–41, doi:10.1016/j.epsl.2010.02.016.

Friedrich, O., Wilson, P.A., Bolton, C.T., Beer, C.J., Schiebel, R. (2013). Late Pliocene to early Pleistocene changes in the North Atlantic Current and suborbital-scale sea-surface temperature variability. Paleoceanography 28. doi:10.1002/palo.20029.

Hennissen, J.A.I.; Head, M.J., De Schepper, S. et al. (2014). Palynological evidence for a southward shift of the North Atlantic Current at similar to 2.6 Ma during the intensification of late Cenozoic Northern Hemisphere glaciation. Paleoceanography 29(6), 564–580. doi:10.1002/2013PA0025423.

Lang, D.C., Bailey, I., Wilson, P.A., Chalk, T.B., Foster, G.L., Gutjahr, M. (2016). Incursions of southern-sourced water into the deep North Atlantic during Late Pliocene glacial intensification. Nature Geoscience. doi:10.1038/ngeo2688.

Mudelsee, M., Raymo, M.E. (2005). Slow dynamics of the Northern Hemisphere glaciation. Paleoceanography 20, PA4022. http://dx.doi.org/10.1029/2005PA001153.

Naafs, B.D.A., Hefter, J., Stein, R. (2014). Dansgaard-Oeschger forcing of sea surface temperature variability in the midlatitude North Atlantic between 500 and 400 ka (MIS 12). Paleoceanography 29, 1024–1030. doi:10.1002/2014PA002697.

Rohling, E.J., Foster, G.L., Grant, K.M., Marino, G., Roberts, A.P., Tamisiea, M.E., Williams, F. (2014). Sea-level and deep-sea-temperature variability over the past 5.3 million years. Nature 508, 477–482.

---

## Referee Comment (RC2) · Anonymous Referee #2 · 12 May 2016

Hodell and Channell present a 3.2 myr long stable isotope and physical properties record at millennial-scale resolution from North Atlantic Site U1308 (reoccupation of ODP 609). The article reviews prior work of the region and it contributes new perspectives due to the high temporal resolution of the data.

I enjoyed reading this manuscript as it provides a succinct review of the pertinent literature coupled with new contributions primarily based on the millennial-scale resolution. There is a lot of information, and the authors manage to convey it in a well written and well organized, easy to read manner. The data and figures are of high quality. The

manuscript conveys the value of long, highly resolved proxy records in paleoclimate research. In sum, I believe the manuscript is an important contribution to the field tying together much North Atlantic research of the past decades.

Line 1139: d18O symbol

In section 2.3: Perhaps the authors should point out to the reader what the temporal resolution is of the 2 cm sampling interval. Line 149 has this info pertaining to the physical properties, and it would be helpful to have it in context for the stable isotopes as well.

Line 256, define natural gamma ray as NGR in parenthesis.

---

## Editor Comment (EC1) · E.L. McClymont (Editor) · 8 Jun 2016

As the authors consider their comments on those of the reviewers, some minor considerations are also outlined here: - I also agree with the comments raised by reviewer 2 regarding age resolution. Do the authors see any relationship between these phase shifts and changes in sedimentation rate (e.g. Line 153?). - Line 117 mentions a hiatus which removed MIS G1 and G2. Does this hiatus have any bearing on the timing or expression of climate shift at 2.7 Ma? - Line 238 should say Ma not ka? - The results section details considerable efforts to understand the evolution of different cyclicity within

the data sets, some of which look quite comparable between different proxies. Did the authors consider investigating whether leads/lags could be determined for their proxies from this site? - Section 4.1 (intensification of northern hemisphere glaciation): How precise are the Balco and Rovey (2010) dates for the two expansions of the Laurentide ice sheet? Could it be that the events onshore are synchronous with those determined from the ocean, within age model error, rather than appearing to lag them (when the ice sheet reaching the sea to provide IRD must have been extensive?) - Paragraph ending line 399: it isn't entirely clear here where the MIS 4 analogy ends and where the data for the time period being investigated starts. Perhaps clarify this 'using MIS 4 as an analogue we infer a larger Laurentide ice sheet compared to Eurasia' - Line 540: 'very large ice and or very cold water'. Does the deep water temperature reconstruction of Sosdian and Rosenthal offer any insights here? - Figure 1 - please ensure that the relevant permissions have been sought for the basemap of this figure, which is taken from Stokes and Clark 2001 (the only 'modifications' to this map are the additions of the site locations)

––––––––––––––––––––––––––––––––––––

---

## Author Comment (AC1) · 4 Jul 2016

We sincerely thank Ian Bailey, Erin McClymont and an anonymous reviewer for their thoughtful comments on our manuscript. Here we respond point-by-point to the questions raised by the reviewers and editor. Reviewer comments are indicated by *italics* and our response is underlined.

**Referee #1 (Ian Bailey)**
Line by line comments

*Line 35: For those less familiar with DSDP/(I)ODP best to spell these acronyms out in full here.*
Yes, agreed.

*Line 37: might help to add a time in Ma in parentheses after 'latest Pliocene'.*
Agreed, will add (2.7 Ma)

*Line 113: Was this modification made by Channell et al. (2016) or was it this study? There seems to be a consistent 9 cm depth offset between the depths assigned to the ages (so for depths greater than ~100 m) presented in Tables 1 of this Ms and of Channell et al. (2016). I apologise if I've got this wrong, but if I've read the tables correctly, does this represent an a slight modification of the Channell et al. (2016) age model in this Ms?*

Well spotted. We uncovered several minor errors in the U1308 splice and the assigned mcds reported by Channell et al. (2016) in QSR. We have corrected these errors in the CP paper and will present a revised splice table and age-depth control points. We will state in the revised paper that the age model is a modification of the one presented in the QSR paper.

*Section 2.3: Do all the benthic $\partial_{18}O$ data come from Hodell et al. (2008) and Channell et al. (2016)?*
Yes, but as noted above, the mcds were miscalculated in Channell et al. (2015). The corrected splice table, isotope data and age model will be attached to this paper.

*Is what you present here for the first time the associated $\partial_{13}C$ data for the older than ~1.5 Ma interval?*
Yes, the d13C data have not been presented previously nor has the bulk carbonate $\delta^{18}O$ records of Sites U1308 (older than 1.5 Ma) and 982.

*If so, you could save text by simply saying you utilise previously published stratigraphies based on benthic $\partial_{18}O$ and present a new benthic $\partial_{13}C$ record from the >1.5 Ma samples analysed by Channell et al. (2016) that extends the previously published $\partial_{13}C$ from Hodell et al. (2008) back to ~3 Ma.*

We will change the text to reflect the fact that we are not using the exact same stratigraphy as published by Channell et al. (2016).

*Your comparison of 607-U1308 stable isotope data in Section 3.1 Ma would benefit*

*from using d₁₈O/d₁₃C splices for 607/U1313 (so using the Bolton et al. (2010)/Lang et al. (2014) data for >2.4 Ma). The U1313 stable isotope records for ~3.3-2.4 Ma are twice the resolution of the 607 record for this time, and using these data will modify some statements you make in this section.*

This is a sensible suggestion and we think it is worthy of an additional figure (shown below).

[Figure]

Comparison of benthic $\delta^{18}$O for Sites U1308 (blue) and U1313 (green) and $\delta^{13}$C for Sites U1308 (black) and U1313 (red).

*Lines 189–190: Is U1308 ∂₁₃C really typically that much more negative than that from U1313 during MIS G6? It's hard to see the detail in your Figure 3, but it looks as though the much more U1308 negative signal can be attributed to two data points.*

The difficulty in seeing the detail in Figure 3 argues for an additional figure that expands the Site U1308 record in comparison to Site U1313 for the interval from 3.2 to 2.4 Ma. The negative $\delta^{13}$C values in G6 is defined by 3 points. All three samples were measured on specimens of *C. wuellerstorfi* (not *C. kullenbergi*) so the low values are not caused by the species analyzed. We have no reason to reject the three low $\delta^{13}$C results in MIS G6.

*Instead it seems that most of the time U1308 ∂₁₃C is only ~0.2-3‰ lighter than at U1313 during G6. This difference may point towards some fundamental difference in source/aging of ∂₁₃C at the deeper U1308 relative to 607. During G6 the ∂₁₃C gradient between U1313 and records of end-member NCW ∂₁₃C (e.g. potentially assessed from Site 982) is still relatively large (Lang et al., 2016). If there is significant SCW at U1313 during MIS G6 in the deep (3.4 km) western North Atlantic, then the lower ∂₁₃C values at the deeper (3.8 km), albeit more northerly eastern basin Site U1308 may reflect that there is a stronger SCW influence at U1308 than at U1313 during MIS G6 (is that really likely?).*

The $\delta^{13}$C record at Site U1308 does seem lower and more variable than U1313 during most glacial stages.  This could be related to a number of possible causes, including both methodological and real effects. Site U1313 was sampled every 10 cm which is equivalent to a temporal sampling resolution of 2-3 kyrs. Site U1308 was sampled every 5 cm which is equivalent to a temporal sampling resolution of 625 yrs.  Thus,

the $\delta^{13}C$ lows at Site U1308 may reflect millennial-scale events that are not captured by the lower resolution record of Site U1313.  Taking a 3-point running mean of the U1308 record to compensate for the differing sampling resolutions yields better agreement but benthic $\delta^{13}C$ is still lower at U1308 than U1313 for some glacial periods.

Isotopic variability is also dependent upon the species and number of benthic specimens used for analysis. Bolton et al. (2010) report that 2 to 8 individuals of *C. wuellerstorfi* were typically analyzed per sample.  We used 1 to 5 individuals of C. wuellerstorfi or *C. kullenbergi*.  In some studies, *C. kull*enbergi has been reported to have lower $\delta^{13}C$ values than *C. wuellerstorfi* because it lives infaunally. We have measured 312 pairs of *C. wuellerstorfi* and *C. kullenbergi* from the same samples at Site U1308 and have not found a consistent offset (see figure below). The number of specimens analyzed can also affect $\delta^{13}C$ variability – in general, we measured slightly fewer specimens at Site U1308 than used for Site U1313

[Figure]

Same figure as above but $\delta^{13}C$ data have been smoothed with a 3-point running average to compensate for the differing sampling resolutions of the two records.

[Figure]

Comparison of paired isotopic measurements of *C. wuellerstorfi* and *C. kulle*nbergi in the same samples from Site U1308.

Excluding methodological explanations, the $\delta^{13}$C differences may reflect real differences at the two sites, which are only 400 m apart in depth. Carbon isotope differences have been report between the eastern and western Atlantic during the last glacial period below the sill separating the basins (Curry and Lohmann, 1983). Decreased advection of deep water into the eastern basin and increased residence time may result in a $\delta^{13}$C difference between eastern and western Atlantic basins.

*Alternatively, the waters bathing U1308 may be dense overflow waters from the north (Bell et al., 2015), although the similar $\partial_{18}O$ values at U1313 and U1308 during this glacial would suggest otherwise.*

Benthic $\delta^{18}$O values are a few tenths of a per mil greater at Site U1308 than U1313. This could reflect lower temperature or greater $\delta^{18}$O of the watermass at Site U1308 than U1313, but might also result from interlaboratory calibration issues. A detailed interlaboratory cross calibration would be necessary before interpreting these differences as significant.

*Line 223–225: Do LGM iceberg drift models of Grant Biggs and Ros D'Eath support the notion that British Chalk/Scandinavian rocks might be a notable source of IRD to U1308? Don't they show it is unlikely that many Scandinavian icebergs/IRD would reach south of Iceland.*

Death et al. (2006) found there are two main collection zones for icebergs; one in the Norwegian–Greenland basin, and one to the southeast of Iceland. The temperature gradient and sub-polar gyre (surface ocean currents) has a strong influence on an iceberg melt once it exits the Norwegian–Greenland basin. Thus, the drift paths are highly dependent upon model parameters chosen. We think it is entirely possible for European icebergs to reach Site U1308; thus, we cannot discount European carbonates as a potential source. In addition, fine-grained detrital carbonate may be more widely dispersed than coarse IRD as it is transported in suspension by ocean currents.

*Line 235: $\partial_{18}O$ (benthic – bulk) increase during MIS 82 is consistent with the fact that this glacial may be characterised by the first late Pleistocene-magnitude sea-level fall Rohling et al. (2014).*

In the revised manuscript, we will note the first 'deep' glaciation described by Rohling et al. (2014) corresponding to MIS 82.

*Line 245: the sentence here reads as though you are saying that there is Ca/Sr data in Figure 4.*

We will reword.

*Line 323: Bailey et al. (2012) is a good reference for North Atlantic IRD sources during MIS 100, but the key reference for evidence of a dominantly Archaean provenance for North Atlantic IRD prior to MIS 100 should be Bailey et al. (2013)*

*where that observation was published for the first time.*

We will reference Bailey et al. (2013)

*Line 330: Raymo et al. (1992) interpret a divergence in Site 607 ∂13C towards values more negative than that of Site 552 during MIS 100 as the first evidence for decreased NCW in the deep North Atlantic Ocean during iNHG. That view has been updated recently in Lang et al. (2016), since it seems that based on U1313 ∂13C and fish debris εNd that MIS G6 is the first glacial associated with significant (and potentially LGM magnitude) SCW incursion into the deep North Atlantic Ocean.*

We had not seen Lang et al. (2016) when we wrote the paper. We will reference this work and update the text to reflect its findings.

*Line 341: Perhaps cite Rohling et al. (2014) here for magnitude of MIS 82 glaciation.*

Will do.

*Lines 342–344: We didn't find sand IRD in U1308 sediments for MIS G6 when studying at a 30 cm sampling resolution (Bailey et al., 2010). Bolton et al. (2010) and Lang et al. (2014) have shown through higher resolution analyses (every 5 cm) that sand IRD is similarly absent in sediments deposited at U1313 until MIS G4, but that values of ~40 grains gram (comparable to the LG scenario at U1313 outside of H-events; Lang et al., 2016) do not occur at this site until MIS 100. These more recent studies have updated the view of when significant icebergs arrived at 40°N based on DSDP studies (Raymo et al., 1986; Kleiven et al., 2002) and support what your data show, i.e. that widespread iceberg rafting and IRD deposition across the North Atlantic Ocean did not occur until MIS 100…reflecting the true large magnitude of that NH glaciation relative to previous cold stages (as potentially also confirmed by, e.g. Balco and Rovey, 2010; Brigham-Grette et al., 2013).*

We will revise the discussion to reflect the significance of the widespread IRD deposition in the North Atlantic beginning with MIS 100.

*Line 345: MIS 94-52 broadly coincides with inference on increased AMOC strength by Bell et al. (2015). Do your eastern basin U1308 ∂13C data support the Bell et al. interpretation, or suggest an alternative origin of the Walvis Ridge 'overflow' signal they report?*

It's difficult to infer transport (e.g., "AMOC strength") from the distribution of a non-conservative nutrient tracer like $\delta^{13}C$ alone. We are skeptical of the interpretation of "maximum AMOC" between 2.0 and 1.5 Ma by Bell et al. (2015). In our view, $\delta^{13}C$ is most useful for reconstructing vertical carbon isotope gradients as a tracer of changes in carbon storage in the deep sea.

*Line 352: the low benthic ∂13C values you report for U1308 in the early Pleistocene should be discussed in the context of the ideas of Bell et al. (2015). You may end up dismissing this suggestion (if you haven't already), but I think this is worth considering because Site 607 doesn't record significant evidence for major shoaling of NCW between 1.5-2 Ma (Lang et al., 2016), and you think it would do if FIS*

*meltwater was impacting significantly on NADW production at this time. The NCW cell can shoal and AMOC can remain relatively strong, but models suggest that if AMOC is reduced then the NCW cell has to shoal. Can we rule out productivity aging of benthic ∂13C at U1308?*

See comment above. We can't rule out ageing of deep water in the eastern basin during glacials. The deep eastern Atlantic is partially isolated from the deep western Atlantic by the Mid-Atlantic Ridge (MAR) and displays higher nutrient concentrations below 3700 m than at the corresponding depths in the western Atlantic.

*Line 368: A obvious question here is "based on the records we've got so far, does it look as though the magnitude and spatial fingerprint of suborbital climate change observed for MIS 3 replicated at any other time during the past ~3 Ma?" The short answer is probably yes, with evidence for DO events as far south as 30°N since ~0.9 Ma (Ferretti et al., 2010; Weirauch et al., 2008). Prior to this time strong evidence exists for DO-like events during MIS 40 and 38 (~1.3 Ma) at 37°N in the northeastern equatorial Atlantic (Birner et al., 2016), but seemingly not at 30°N in the northwestern North Atlantic (Weirauch et al., 2008). No convincing evidence exists anywhere yet for DO–magnitude change during any earliest Pleistocene glacials, e.g. during MIS 100 (one of the more well studied cold stages for this time), but instead muted suborbital change in planktic ∂18O and SST ~40-60°N (Bartoli et al., 2006; Becker et al., 2006; Bolton et al., 2010; Friedrich et al., 2013).*

The short answer is we need more long records of millennial variability to determine the regional fingerprint of the signal in the North Atlantic. We will stress that our conclusions about IRD and millennial variability apply to Site U1308 in the central North Atlantic only – cores from other regions may have a different expression of millennial climate change.

*Benthic ∂18O records suggest our planet's climate system has been crossing this +3.5‰ threshold during glacials ever since ~2.7-2.5 Ma. If we assume that this benthic ∂18O value corresponds to a relatively narrow range of NH ice-sheet growth then the available evidence suggests that the spatial fingerprint of DO-like change over the past 3 Ma is not consistent with the climate system responding in a repeatable (pseudo predictable) manner to it sitting in an intermediate ice-volume window. If it did, we should expect to see the same spatial pattern more or less emerging for amplification of suborbital climate change during all big benthic ∂18O glacials (>+3.5‰) from ~2.5 Ma. The occurrence of DO-like change is clearly linked to NH ice sheet size, but records of the 41-kyr world suggest to me it is too simplistic to think of it as a straight forward ice-volume feedback (or our understanding of NH ice sheet volume during the 41-kyr world needs revision). If as yet undiscovered DO-like magnitude change really is restricted to the highest latitudes during the earliest Pleistocene, then that suborbital change (and the mechanisms responsible for it) do not seem to be analogous to events during the LG.*

Our view of millennial variability is strongly (mis)shaped by the last glacial period

that, as you suggest, may not be representative of older glacial periods, particularly those of the 41-kyr world.  We agree that millennial variability in the 41-kyr world was not entirely analogous to MIS 3; for example, there were no Heinrich events prior to 650 ka.  We will clarify the text to reflect this point.  We also agree the relationship between the +3.5‰ threshold and millennial variability is likely not straightforward.  We suggest the critical factor is how ice growth affects the volume, rate, and location of freshwater discharge to the North Atlantic Ocean relative to the source areas of deepwater formation.  In this regard, the importance of the European ice sheet may be underestimated especially for explaining millennial variability during the period of glacial onset.

*Line 375–278: McIntye et al. (2001) present strong evidence for millennial-scale changes in iceberg rafting to Site 983 in the early Pleistocene (~1.93-1.75 Ma). You also know we found the same thing at U1313 and U1308 during the much older MIS 100 (Bolton et al., 2010; Bailey et al., 2010) and at U1308 during MIS G4 (Bailey et al., 2010), but none of these earliest Pleistocene events are yet found to be associated with large amplitude swings in SST/$\partial_{18}O$ (Becker et al., 2006; Bartoli et al., 2006; Bolton et al., 2010; Friedrich et al., 2013). The point I am trying to make here, and one you obviously appreciate, is that millennial-scale pulses of IRD deposition do not necessarily imply large magnitude swings in climate on such timescales, just that there are likely millennial-scale swings in climate driving the mass balance of ice-sheets/glacier at the coast at those times.*

We agree that the mere occurrence of IRD at a particular site doesn't necessarily imply there was a large climate response to the event. This is a classic "chicken and the egg problem" -- i.e., the degree to which iceberg discharge is the cause of climate change versus the consequence of stadial conditions (as recently discussed by Barker et al, 2015).  Likewise, the absence of IRD at a single site does not necessarily preclude freshwater forcing elsewhere. We need other long records of IRD and millennial variability similar to Site U1308 to properly evaluate the magnitude and spatial variability of  millennial variability beyond the last glacial period. We will add this caveat to the paper.

*Marshall and Koutnik (2006) show that millennial-scale episodes of iceberg rafting can still be anticipated with muted suborbital climatic variability, but that such pulses might be set against a steadier background of IRD inputs, making them less distinct in the sediment record. If suborbital change during the earliest Pleistocene was muted relative to the late Pleistocene, we may therefore find that overall IRD inputs during earliest Pleistocene glacials were higher, but that suborbital-scale IRD pulses superimposed on this signal were muted, relative to IRD inputs during e.g. MIS 3 at U1308. Maybe it is best to look for this at a site further north where the iceberg/IRD survivability issue less strongly influences IRD inputs, but maybe worth thinking along these lines here since your record is the only suborbital proxy IRD record we have that spans the entire Quaternary.*

Marshall and Koutnik (2006) distinguished Heinrich events from "background" IRD making the point that they represent different glacial processes – i.e., dynamic (surging) versus mass balance processes, respectively.  The first Heinrich event and presumably the dynamic glacial processes responsible first occurred at 650 ka in MIS

16. In the paper, we suggest the IRD events prior to MIS 16 were more similar to the "background IRD" of MIS 3. As you suggest, the "background" IRD events may have a more muted climate response. We will add this point to the discussion.

*Line 384/436/561: is Figure 5 the correct figure to cite here? Don't you mean Fig. 6 evol. power spec?*

Yes, the figure call should be for Fig. 6 (not 5).

*Line 391: please place a horizontal line at the benthic $\partial_{18}O$ value of +4 ‰ (~MIS 4) and +3.5 ‰ (McManus) to guide the reader's eye when they examine Figure 4.*

Will do.

*Line 397: 'ice volume was about twice as great in North America compared to Eurasia'. I don't disagree that your datasets suggest that the deposition of HS-sourced material increased from 1.6 Ma (seems consistent with U1313 data from Naafs et al., 2013), but how do you then extend that to what seems like a relatively precise quantification of relative differences in ice volume?*

We are referring to MIS 4 here but I agree it sounds like it refers to the post-1.5Ma period. We will make this point clear.

*Line 410: Good to plot an indicator of IRD in Figure 9 to help the reader see more easily the relationship between iceberg rafting to U1308 and U1313 and the SST gradient evolution.*

Will do.

*Lines 412–426: again, see the recent findings of Lang et al. (2016) for new context on the pioneering observations of Raymo et al. (1990; 1992) and those made subsequently by e.g. Lisiecki (2014).*

We will reference Lang et al. (2016) and update the text to reflect the findings of this study.

*Lines 446–462: Have you compared your bulk $\partial_{18}O$ record and/or $\partial_{18}O$ (benthic – bulk) to Steve Barker's synthetic Greenland DO record? Is it may be worth showing a plot of this, if only in the supplementary guide. How does the variability in your record(s) for MIS 41-37 compare to those from your work in Birner et al. (2016)? Do we see evidence for the same number of ice-rafting events at U1308 as reported by Raymo et al. (1998) further north at ODP Site 983 during MIS 40 that you've tied convincingly to the DO-like variability seen in G. bulloides $\partial_{18}O$ from the Iberian Margin?*

There are too many millennial "events" in the Barker synthetic record for a meaningful comparison (see figure below). As discussed above, not every cooling is necessarily associated with an IRD event. In addition, the Greenland synthetic is a

derived record based on the assumption of a bipolar seesaw between Antarctica and Greenland; thus, it doesn't necessarily exactly reproduce Greenland or North Atlantic climate. There's much better agreement between IRD at Site 983 (Barker et al., 2015) and bulk carbonate δ[18]O at Site U1308. For the earlier Pleistocene, Birner et al. (2016, Fig. 7) showed reasonable agreement between millennial variability (particularly the larger stadial events) at Sites U1308 and U1385 (Iberian Margin) for MIS 38 and 40.

[Figure]

Comparison of the Greenland synthetic record of Barker et al. (2011) and bulk d18O at

[Figure]

Comparison of IRD at Site 983 (Barker et al. 2015) and bulk d18O at Site U1308.

*Lines 485–486: what's the Site 982 bulk ∂18O data source? Are these data produced for this study? If so, please mention these analyses in your methods text. If not, the data source needs including.*

These are new data not reported previously. Methods will be updated.

*Figures*

*Figure 1. Nice map. Perhaps state what the yellow/green triangles mean in your key*

*too.*
Will do

*Figure 2. I think it would still help to have a key on the figure so it is easier for the reader to work out which record is the LR04 vs U1308 ∂18Ob (like you do in Figure 10).*
Will do

*Figure 3. A key showing which records are from 607 versus U1308 would aid the reader. I suggest labelling the horizontal lines with '21', '41' and '100' kyr. Ditto Figures 4, 6 and 7.*
Will do.

*Figure 4. Please add the horizontal lines for MIS 5b, 4 and 2 onto the benthic ∂18O data (as it is on Figure 3). Please also label the key bulk carbonate ∂18O values referred to the in text, e.g. the -4 ‰ value characteristic of H-layers and the -2 ‰ value characteristic of DO-type ice-rafting events.*
Will do

*Figure 6. Given the density increase with depth, to make the suborbital events even clearer, it might be helpful to detrend the density data plotted in Fig. 6 by subtracting the linear best fit from it.*
One would want to subtract out the downcore increase in density due to sediment compaction but I don't think subtracting a linear best fit accomplishes this task (see figure below). We prefer to leave the figure as is.

[Figure]

*Maybe combine Figures 8 and 9 to help the reader see clearly how the 982-U1313 SST gradient evolves alongside changes in IRD inputs to these two sites.*
Yes, good suggestion.

*Figure 13. Please label site names on dust records. Could also do with labelling key MIS on the LR04 or including vertical guide lines. It would also help to label all HS H-layers on the relevant figures to help tell apart HS-sourced H-layers and non-H-event (DO) IRD deposition in your bulk ∂18O record.*
Will do.

*Figure 12: Data sources for U1304 NGR and benthic ∂₁₈O not given in caption, or is it all presented in Xuan et al. (submitted)? If so a quick revision of the caption text is needed.*
Xuan et al. (2016) can be cited as the paper is now published in EPSL.

*Additional references cited:*
Will add

**Anonymous Referee #2**

*Line 1139: d18O symbol*
Fixed.

*In section 2.3: Perhaps the authors should point out to the reader what the temporal resolution is of the 2 cm sampling interval. Line 149 has this info pertaining to the physical properties, and it would be helpful to have it in context for the stable isotopes as well.*
Agreed. This change will be instituted.

*Line 256, define natural gamma ray as NGR in parenthesis*
Agreed

**Editors Comments:**

*Do the authors see any relationship between these phase shifts and changes in sedimentation rate (e.g. Line 153?)?*

No, there is no apparent relationship between the times of mode transitions and sedimentation rate (see figure).

[Figure]

Age-depth and interval sedimentation rates for Site U1308.

*Line 117 mentions a hiatus which removed MIS G1 and G2. Does this hiatus have any bearing on the timing or expression of climate shift at 2.7 Ma?*

The duration of the hiatus is short (50 kyrs between 2.65 and 2.5 Ma) and thus it does not significantly alter the expression of the climate shift at 2.7 Ma. The hiatus (loss of section ) is represented by evidence of slumping at Site U1308 in the vicinity of 197 mcd (see Fig. 3 and discussion of Channell et al., 2015).

*Line 238 should say Ma not ka?*
Fixed

*The results section details considerable efforts to understand the evolution of different cyclicity within the data sets, some of which look quite comparable between different proxies. Did the authors consider investigating whether leads/lags could be determined for their proxies from this site?*

We did indeed produce cross wavelet plots of the various parameters against one another and relative to orbital forcing (etp) to examine the evolution of phase and coherency over the past 3200 ka. An example of the cross wavelet between benthic $\delta^{18}O$ and $-\delta^{13}C$ is shown below. The manuscript was becoming quite long and we didn't feel the cross wavelet analysis added a lot of additional insight to the evolution of Quaternary climate. However, we would be willing to include the cross wavelet plots in the supplement if they are deemed worthwhile.

[Figure]

Cross wavelet of δ¹⁸O and negative δ¹³C (multiplied by -1). Arrows indicate phase with in-phase pointng to right and anti-phase to the left. Benthic δ¹⁸O leading -δ¹³C by 90° would point straight down.

*Section 4.1 (intensification of northern hemisphere glaciation): How precise are the Balco and Rovey (2010) dates for the two expansions of the Laurentide ice sheet? Could it be that the events onshore are synchronous with those determined from the ocean, within age model error, rather than appearing to lag them (when the ice sheet reaching the sea to provide IRD must have been extensive?)*

The error on the age of Balco and Rovey (2010) is 2.421 +/- 0.143 Ma , giving a range of 2.298 to 2.564 Ma so, yes, the age could coincide with MIS 100 (2.52 Ma) and the younger MIS 98 and 96.

*Paragraph ending line 399: it isn't entirely clear here where the MIS 4 analogy ends and where the data for the time period being investigated starts. Perhaps clarify this 'using MIS 4 as an analogue we infer a larger Laurentide ice sheet compared to Eurasia'*

Yes, we agree and will clarify this point.

*Line 540: 'very large ice and or very cold water'. Does the deep water temperature reconstruction of Sosdian and Rosenthal offer any insights here?*

One has to be cautious with the deep-water temperature record of Sodian and Rosenthal (2009) because Mg/Ca was measured on the epibenthic *C. wuellerstorfi* that may be affected

by variations in carbonate ion (see comment by Yu and Brocecker, 2010).

*Figure 1 - please ensure that the relevant permissions have been sought for the basemap of this figure, which is taken from Stokes and Clark 2001 (the only 'modifications' to this map are the additions of the site locations)*

Copyright clearance to reproduce Figure 1 has been sought and granted by Elsevier through the Copyright Clearance Center's RightsLink service.

---

## Author Response (AR1)

Here we respond point-by-point to the questions raised by the reviewers and editor. Reviewer comments are indicated by *italics* and our response is underlined.

**Referee #1 (Ian Bailey)**
Line by line comments

*Line 35: For those less familiar with DSDP/(I)ODP best to spell these acronyms out in full here.*
The acronyms have been spelled out..

*Line 37: might help to add a time in Ma in parentheses after 'latest Pliocene'.*
(2.7 Ma) has been added.

*Line 113: Was this modification made by Channell et al. (2016) or was it this study? There seems to be a consistent 9 cm depth offset between the depths assigned to the ages (so for depths greater than ~100 m) presented in Tables 1 of this Ms and of Channell et al. (2016). I apologise if I've got this wrong, but if I've read the tables correctly, does this represent an a slight modification of the Channell et al. (2016) age model in this Ms?*

We uncovered several minor errors in the U1308 splice and the assigned mcds reported by Channell et al. (2016) in QSR. We have corrected these errors in the CP paper and present a revised splice table (Table S1) and age-depth control points (Table S2). We state in the paper that the age model is a modification of the one presented in the QSR paper.

*Section 2.3: Do all the benthic $\partial_{18}O$ data come from Hodell et al. (2008) and Channell et al. (2016)?*
Yes, but as noted above, the mcds were miscalculated in Channell et al. (2016). The corrected splice table, isotope data and age model are attached to this paper.

*Is what you present here for the first time the associated $\partial_{13}C$ data for the older than ~1.5 Ma interval?*
Yes, in addition to the bulk carbonate $\delta^{18}O$ records of Sites U1308 and 982, which are not published in Channell et al. (2016).

*If so, you could save text by simply saying you utilise previously published stratigraphies based on benthic $\partial_{18}O$ and present a new benthic $\partial_{13}C$ record from the >1.5 Ma samples analysed by Channell et al. (2016) that extends the previously published $\partial_{13}C$ from Hodell et al. (2008) back to ~3 Ma.*

We changed the text to reflect the fact that we are not using the exact same stratigraphy as published by Channell et al. (2016).

*Your comparison of 607-U1308 stable isotope data in Section 3.1 Ma would benefit from using d18O/d13C splices for 607/U1313 (so using the Bolton et al. (2010)/Lang et al. (2014) data for >2.4 Ma). The U1313 stable isotope records for ~3.3-2.4 Ma are twice the resolution of the 607 record for this time, and using these data will modify*

*some statements you make in this section.*

We added an additional figure (Fig 4) that compares benthic $\delta^{18}O$ for Sites U1308 (blue) and U1313 and $\delta^{13}C$ for Sites U1308 and U1313.

*Lines 189–190: Is U1308 ∂13C really typically that much more negative than that from U1313 during MIS G6? It's hard to see the detail in your Figure 3, but it looks as though the much more U1308 negative signal can be attributed to two data points.*

The difficulty in seeing the detail in Figure 3 argues for an additional figure that expands the Site U1308 record in comparison to Site U1313 for the interval from 3.2 to 2.4 Ma (this is new Fig. 4). The negative $\delta^{13}C$ values in G6 is defined by 3 points. All three samples were measured on specimens of *C. wuellerstorfi* (not *C. kullenbergi*) so the low values are not caused by the species analyzed. We have no reason to reject the three low results in MIS G6.

*Instead it seems that most of the time U1308 ∂13C is only ~0.2-3‰ lighter than at U1313 during G6. This difference may point towards some fundamental difference in source/aging of ∂13C at the deeper U1308 relative to 607. During G6 the ∂13C gradient between U1313 and records of end-member NCW ∂13C (e.g. potentially assessed from Site 982) is still relatively large (Lang et al., 2016). If there is significant SCW at U1313 during MIS G6 in the deep (3.4 km) western North Atlantic, then the lower ∂13C values at the deeper (3.8 km), albeit more northerly eastern basin Site U1308 may reflect that there is a stronger SCW influence at U1308 than at U1313 during MIS G6 (is that really likely?).*

The $\delta^{13}C$ record at Site U1308 does seem lower and more variable than U1313 during most glacial stages.  This could be related to a number of possible causes, including both methodological and real effects. Site U1313 was sampled every 10 cm which is equivalent to a temporal sampling resolution of 2-3 kyrs. Site U1308 was sampled every 5 cm which is equivalent to a temporal sampling resolution of 625 yrs.  Thus, the $\delta^{13}C$ lows at Site U1308 may reflect millennial-scale events that are not captured by the lower resolution record of Site U1313.  These possibilities are discussed in the text.

Isotopic variability is also dependent upon the species and number of benthic specimens used for analysis. Bolton et al. (2010) report that 2 to 8 individuals of *C. wuellerstorfi* were typically analyzed per sample.  We used 1 to 5 individuals of C. wuellerstorfi or *C. kullenbergi*.  In some studies, *C. kull*enbergi has been reported to have lower $\delta^{13}C$ values than *C. wuellerstorfi* because it lives infaunally. We have measured 312 pairs of *C. wuellerstorfi* and *C. kullenbergi* from the same samples at Site U1308 and have not found a consistent offset. We have added an new figure to the supplement (Fig. S1) that shows this comparision. The number of specimens analyzed can also affect $\delta^{13}C$ variability – in general, we measured slightly fewer specimens per sample at Site U1308 than used for Site U1313

Excluding methodological explanations, the $\delta^{13}C$ differences may reflect real differences at the two sites, which are only 400 m different in water depth. Carbon isotope differences have been reported between the eastern and western Atlantic during the last glacial period below the sill separating the basins (Curry and Lohmann, 1983). Decreased advection of deep water into the eastern basin and increased residence time may result in a $\delta^{13}C$ difference between eastern and western basins. This is now mentioned in the text.

*Alternatively, the waters bathing U1308 may be dense overflow waters from the north (Bell et al., 2015), although the similar $\partial_{18}O$ values at U1313 and U1308 during this glacial would suggest otherwise.*

Benthic $\delta^{18}O$ values are a few tenths of a per mil greater at Site U1308 than U1313. This could reflect lower temperature or greater $\delta^{18}O$ of the watermass at Site U1308 than U1313, but might also result from interlaboratory calibration issues. A detailed interlaboratory cross calibration would be necessary before interpreting these differences as significant.

*Line 223–225: Do LGM iceberg drift models of Grant Biggs and Ros D'Eath support the notion that British Chalk/Scandinavian rocks might be a notable source of IRD to U1308? Don't they show it is unlikely that many Scandinavian icebergs/IRD would reach south of Iceland.*

D'Eath et al. (2006) found there are two main collection zones for icebergs; one in the Norwegian–Greenland basin, and one to the southeast of Iceland. The temperature gradient and sub-polar gyre (surface ocean currents) has a strong influence on an iceberg melt once it exits the Norwegian–Greenland basin. Thus, the drift paths are highly dependent upon model parameters chosen. I think it is entirely possible for European icebergs to reach Site U1308; thus, we should not discount European carbonates as a potential source.

*Line 235: $\partial_{18}O$ (benthic – bulk) increase during MIS 82 is consistent with the fact that this glacial may be characterised by the first late Pleistocene-magnitude sea-level fall Rohling et al. (2014).*

In the revised manuscript, we have noted the first 'deep' glaciation described by Rohling et al. (2014) corresponds to MIS 82.

*Line 245: the sentence here reads as though you are saying that there is Ca/Sr data in Figure 4.*

The sentence has been reworded.

*Line 323: Bailey et al. (2012) is a good reference for North Atlantic IRD sources during MIS 100, but the key reference for evidence of a dominantly Archaean provenance for North Atlantic IRD prior to MIS 100 should be Bailey et al. (2013) where that observation was published for the first time.*

The reference has been changed to Bailey et al. (2013)

*Line 330: Raymo et al. (1992) interpret a divergence in Site 607 ∂13C towards values more negative than that of Site 552 during MIS 100 as the first evidence for decreased NCW in the deep North Atlantic Ocean during iNHG. That view has been updated recently in Lang et al. (2016), since it seems that based on U1313 ∂13C and fish debris εNd that MIS G6 is the first glacial associated with significant (and potentially LGM magnitude) SCW incursion into the deep North Atlantic Ocean.*

We had not seen Lang et al. (2016) when we wrote the paper.  We now reference this work and have updated the text to reflect its findings.

*Line 341: Perhaps cite Rohling et al. (2014) here for magnitude of MIS 82 glaciation.*

*Rohling et al. (2014 )is now cited..*

*Lines 342–344: We didn't find sand IRD in U1308 sediments for MIS G6 when studying at a 30 cm sampling resolution (Bailey et al., 2010). Bolton et al. (2010) and Lang et al. (2014) have shown through higher resolution analyses (every 5 cm) that sand IRD is similarly absent in sediments deposited at U1313 until MIS G4, but that values of ~40 grains gram (comparable to the LG scenario at U1313 outside of H-events; Lang et al., 2016) do not occur at this site until MIS 100. These more recent studies have updated the view of when significant icebergs arrived at 40°N based on DSDP studies (Raymo et al., 1986; Kleiven et al., 2002) and support what your data show, i.e. that widespread iceberg rafting and IRD deposition across the North Atlantic Ocean did not occur until MIS 100...reflecting the true large magnitude of that NH glaciation relative to previous cold stages (as potentially also confirmed by, e.g. Balco and Rovey, 2010; Brigham-Grette et al., 2013).*

We revised the discussion to reflect the significance of the widespread IRD deposition in the North Atlantic beginning with MIS 100.

*Line 345: MIS 94-52 broadly coincides with inference on increased AMOC strength by Bell et al. (2015). Do your eastern basin U1308 ∂13C data support the Bell et al. interpretation, or suggest an alternative origin of the Walvis Ridge 'overflow' signal they report?*

It's difficult to infer transport (e.g., "AMOC strength") from the distribution of a non-conservative nutrient tracer like $\delta^{13}C$ alone.  We are skeptical of the interpretation of "maximum AMOC" between 2.0 and 1.5 Ma by Bell et al. (2015). In our view, $\delta^{13}C$ is most useful for reconstructing vertical carbon isotope gradients as a tracer of changes in carbon storage in the deep sea. None the less, we have mentioned the interpretation of Bell et al. (2015) in the discussion.

*Line 352: the low benthic ∂13C values you report for U1308 in the early Pleistocene should be discussed in the context of the ideas of Bell et al. (2015). You may end up dismissing this suggestion (if you haven't already), but I think this is worth considering because Site 607 doesn't record significant evidence for major shoaling of NCW between 1.5-2 Ma (Lang et al., 2016), and you think it would do if FIS*

*meltwater was impacting significantly on NADW production at this time. The NCW cell can shoal and AMOC can remain relatively strong, but models suggest that if AMOC is reduced then the NCW cell has to shoal. Can we rule out productivity aging of benthic $\partial_{13}C$ at U1308?*

See comment above. I don't think we can rule out ageing of deep water in the eastern basin during glacials. The deep eastern Atlantic is partially isolated from the deep western Atlantic by the Mid-Atlantic Ridge (MAR). For example, the eastern Atlantic below 3700 m displays higher nutrient concentrations than at the corresponding depths in the western Atlantic.

*Line 368: A obvious question here is "based on the records we've got so far, does it look as though the magnitude and spatial fingerprint of suborbital climate change observed for MIS 3 replicated at any other time during the past ~3 Ma?" The short answer is probably yes, with evidence for DO events as far south as 30°N since ~0.9 Ma (Ferretti et al., 2010; Weirauch et al., 2008). Prior to this time strong evidence exists for DO-like events during MIS 40 and 38 (~1.3 Ma) at 37°N in the northeastern equatorial Atlantic (Birner et al., 2016), but seemingly not at 30°N in the northwestern North Atlantic (Weirauch et al., 2008). No convincing evidence exists anywhere yet for DO–magnitude change during any earliest Pleistocene glacials, e.g. during MIS 100 (one of the more well studied cold stages for this time), but instead muted suborbital change in planktic $\partial_{18}O$ and SST ~40-60°N (Bartoli et al., 2006; Becker et al., 2006; Bolton et al., 2010; Friedrich et al., 2013).*

The short answer is we need more long records of millennial variability to determine the regional fingerprint of the signal in the North Atlantic. We have stressed that our conclusions about IRD and millennial variability apply to Site U1308 in the central North Atlantic only – cores from other regions may have a different expression.

*Benthic $\partial_{18}O$ records suggest our planet's climate system has been crossing this +3.5‰ threshold during glacials ever since ~2.7-2.5 Ma. If we assume that this benthic $\partial_{18}O$ value corresponds to a relatively narrow range of NH ice-sheet growth then the available evidence suggests that the spatial fingerprint of DO-like change over the past 3 Ma is not consistent with the climate system responding in a repeatable (pseudo predictable) manner to it sitting in an intermediate ice-volume window. If it did, we should expect to see the same spatial pattern more or less emerging for amplification of suborbital climate change during all big benthic $\partial_{18}O$ glacials (>+3.5‰) from ~2.5 Ma. The occurrence of DO-like change is clearly linked to NH ice sheet size, but records of the 41-kyr world suggest to me it is too simplistic to think of it as a straight forward ice-volume feedback (or our understanding of NH ice sheet volume during the 41-kyr world needs revision). If as yet undiscovered DO-like magnitude change really is restricted to the highest latitudes during the earliest Pleistocene, then that suborbital change (and the mechanisms responsible for it) do not seem to be analogous to events during the LG.*

Our view of millennial variability is strongly (mis)shaped by the last glacial period which, as you suggest, may not be representative of older glacial periods, particularly those of the 41-kyr world. We agree that millennial variability in the 41-kyr world

was not entirely analogous to MIS 3; for example, there were no Heinrich events prior to 650 ka.  We modified the text to reflect this clarification.  We also agree the relationship between the +3.5‰ threshold and millennial variability is likely not straightforward. We suggest the critical factor is how ice growth affects the volume, rate, and location of freshwater discharge to the North Atlantic Ocean relative to the source areas of deepwater formation.  In this regard, the importance of the European ice sheet has been underestimated especially for explaining millennial variability during the period of glacial onset.

*Line 375–278: McIntye et al. (2001) present strong evidence for millennial-scale changes in iceberg rafting to Site 983 in the early Pleistocene (~1.93-1.75 Ma). You also know we found the same thing at U1313 and U1308 during the much older MIS 100 (Bolton et al., 2010; Bailey et al., 2010) and at U1308 during MIS G4 (Bailey et al., 2010), but none of these earliest Pleistocene events are yet found to be associated with large amplitude swings in SST/∂₁₈O (Becker et al., 2006; Bartoli et al., 2006; Bolton et al., 2010; Friedrich et al., 2013). The point I am trying to make here, and one you obviously appreciate, is that millennial-scale pulses of IRD deposition do not necessarily imply large magnitude swings in climate on such timescales, just that there are likely millennial-scale swings in climate driving the mass balance of ice-sheets/glacier at the coast at those times.*

We agree that the mere occurrence of IRD at a particular site doesn't necessarily imply there was a large climate response to the event. This is a classic "chicken and the egg problem" -- i.e., the degree to which iceberg discharge is the cause of climate change versus the consequence of stadial conditions (as recently discussed by Barker et al, 2015).  Likewise, the absence of IRD at a single site does not necessarily preclude freshwater forcing elsewhere. We need other long records of IRD and millennial variability similar to Site U1308 to properly evaluate the magnitude and spatial variability of  millennial variability beyond the last glacial period. We have added this caveat to the paper.

*Marshall and Koutnik (2006) show that millennial-scale episodes of iceberg rafting can still be anticipated with muted suborbital climatic variability, but that such pulses might be set against a steadier background of IRD inputs, making them less distinct in the sediment record. If suborbital change during the earliest Pleistocene was muted relative to the late Pleistocene, we may therefore find that overall IRD inputs during earliest Pleistocene glacials were higher, but that suborbital-scale IRD pulses superimposed on this signal were muted, relative to IRD inputs during e.g. MIS 3 at U1308. Maybe it is best to look for this at a site further north where the iceberg/IRD survivability issue less strongly influences IRD inputs, but maybe worth thinking along these lines here since your record is the only suborbital proxy IRD record we have that spans the entire Quaternary.*

Marshall and Koutnik (2006) distinguished Heinrich events from "background" IRD making the point that they represent different glacial processes – i.e., dynamic (surging) versus mass balance processes, respectively.  The first Heinrich event and presumably the dynamic glacial processes responsible first occurred at 650 ka in MIS 16. In the paper, we suggest the IRD events prior to MIS 16 were more similar to the "background IRD" of MIS 3.  As you suggest, the "background" IRD events have a

more mute climate response. We have added this point to the discussion.

*Line 384/436/561: is Figure 5 the correct figure to cite here? Don't you mean Fig. 6 evol. power spec?*

The figure call has been corrected.

*Line 391: please place a horizontal line at the benthic ∂18O value of +4 ‰ (~MIS 4) and +3.5 ‰ (McManus) to guide the reader's eye when they examine Figure 4.*

Done.

*Line 397: 'ice volume was about twice as great in North America compared to Eurasia'. I don't disagree that your datasets suggest that the deposition of HS-sourced material increased from 1.6 Ma (seems consistent with U1313 data from Naafs et al., 2013), but how do you then extend that to what seems like a relatively precise quantification of relative differences in ice volume?*

We are referring to MIS 4 here but I agree it sounds like it refers to the post-1.5Ma period. We have changes the text to make this clear.

*Line 410: Good to plot an indicator of IRD in Figure 9 to help the reader see more easily the relationship between iceberg rafting to U1308 and U1313 and the SST gradient evolution.*

Done.

*Lines 412–426: again, see the recent findings of Lang et al. (2016) for new context on the pioneering observations of Raymo et al. (1990; 1992) and those made subsequently by e.g. Lisiecki (2014).*

We have referenced Lang et al. (2016) and updated the text to reflect the findings of this study.

*Lines 446–462: Have you compared your bulk ∂18O record and/or ∂18O (benthic – bulk) to Steve Barker's synthetic Greenland DO record? Is it may be worth showing a plot of this, if only in the supplementary guide. How does the variability in your record(s) for MIS 41-37 compare to those from your work in Birner et al. (2016)? Do we see evidence for the same number of ice-rafting events at U1308 as reported by Raymo et al. (1998) further north at ODP Site 983 during MIS 40 that you've tied convincingly to the DO-like variability seen in G. bulloides ∂18O from the Iberian Margin?*

There are too many millennial "events" in the Barker synthetic record for a meaningful comparison (see figure below). As discussed above, not every cooling is necessarily associated with an IRD event. In addition, the Greenland synthetic is a derived record based on the assumption of a bipolar seesaw between Antarctica and Greenland; thus, it doesn't necessarily exactly reproduce Greenland or North Atlantic

climate. There's much better agreement between IRD at Site 983 (Barker et al., 2015) and bulk carbonate δ$^{18}$O at Site U1308. For the earlier Pleistocene, Birner et al. (2016, Fig. 7) showed reasonable agreement between millennial variability (particularly the larger stadial events) at Sites U1308 and U1385 (Iberian Margin) for MIS 38 and 40.

*Lines 485–486: what's the Site 982 bulk ∂$_{18}$O data source? Are these data produced for this study? If so, please mention these analyses in your methods text. If not, the data source needs including.*

These are new data not reported previously. Methods have been updated.

*Figures*

*Figure 1. Nice map. Perhaps state what the yellow/green triangles mean in your key too.*
Done

*Figure 2. I think it would still help to have a key on the figure so it is easier for the reader to work out which record is the LR04 vs U1308 ∂$_{18}$O$_b$ (like you do in Figure 10).*
Done

*Figure 3. A key showing which records are from 607 versus U1308 would aid the reader. I suggest labelling the horizontal lines with '21', '41' and '100' kyr. Ditto Figures 4, 6 and 7.*
Done

*Figure 4. Please add the horizontal lines for MIS 5b, 4 and 2 onto the benthic ∂$_{18}$O data (as it is on Figure 3). Please also label the key bulk carbonate ∂$_{18}$O values referred to the in text, e.g. the -4 ‰ value characteristic of H-layers and the -2 ‰ value characteristic of DO-type ice-rafting events.*
Done

*Figure 6. Given the density increase with depth, to make the suborbital events even clearer, it might be helpful to detrend the density data plotted in Fig. 6 by subtracting the linear best fit from it.*

One would want to subtract out the downcore increase in density due to sediment compaction but I don't think subtracting a linear best fit accomplishes much (see figure below). We prefer to leave the figure as is.

*Maybe combine Figures 8 and 9 to help the reader see clearly how the 982-U1313 SST gradient evolves alongside changes in IRD inputs to these two sites.*
Yes, good suggestion.

*Figure 13. Please label site names on dust records. Could also do with labelling key MIS on the LR04 or including vertical guide lines. It would also help to label all HS H-layers on the relevant figures to help tell apart HS-sourced H-layers and non-H-event (DO) IRD deposition in your bulk ∂18O record.*
Done

*Figure 12: Data sources for U1304 NGR and benthic ∂18O not given in caption, or is it all presented in Xuan et al. (submitted)? If so a quick revision of the caption text is needed.*

Xuan et al. (2016) has been cited as the paper is now published in QSR.

*Additional references cited:*
Added

**Anonymous Referee #2**

*Line 1139: d18O symbol*
Fixed.

*In section 2.3: Perhaps the authors should point out to the reader what the temporal resolution is of the 2 cm sampling interval. Line 149 has this info pertaining to the physical properties, and it would be helpful to have it in context for the stable isotopes as well.*
Agreed. This change has been instituted.

*Line 256, define natural gamma ray as NGR in parenthesis*
Done

**Editors Comments:**

*Do the authors see any relationship between these phase shifts and changes in sedimentation rate (e.g. Line 153?)?*

No, there is no apparent relationship between the times of mode transitions and sedimentation rate. Sedimentation rate figure has been added to supplementary materials.

*Line 117 mentions a hiatus which removed MIS G1 and G2. Does this hiatus have any bearing on the timing or expression of climate shift at 2.7 Ma?*

The duration of the hiatus is short (50 kyrs between 2.65 and 2.5 Ma) and thus it does not significantly affect the expression of the climate shift at 2.7 Ma. The hiatus (loss of section ) is represented by evidence of slumping at Site U1308 in the vicinity of 197 mcd (see Fig. 3 and discussion of Channell et al., 2016).

*Line 238 should say Ma not ka?*
Fixed

*The results section details considerable efforts to understand the evolution of different cyclicity within the data sets, some of which look quite comparable between different proxies. Did the authors consider investigating whether leads/lags could be determined for their proxies from this site?*

We did indeed produce cross wavelet plots of the various parameters against one another and relative to orbital forcing (etp) to examine the evolution of phase and coherency over the past 3200 ka. An example of the cross wavelet between benthic d18O and –d13C is shown below. The manuscript was becoming quite long and we didn't feel the cross wavelet analysis added a lot of additional insight to the evolution of Quaternary climate.

*Section 4.1 (intensification of northern hemisphere glaciation): How precise are the Balco and Rovey (2010) dates for the two expansions of the Laurentide ice sheet? Could it be that the events onshore are synchronous with those determined from the ocean, within age model error, rather than appearing to lag them (when the ice sheet reaching the sea to provide IRD must have been extensive?)*

The error on the age of Balco and Rovey (2010) is 2.421 +/- 0.143 Ma , giving a range of 2.298 to 2.564 Ma so, yes, the age could coincide with MIS 100 (2.52 Ma) and the younger MIS 98 and 96. We've added the error reported by Balco and Rovey (2010) to the text.

*Paragraph ending line 399: it isn't entirely clear here where the MIS 4 analogy ends and where the data for the time period being investigated starts. Perhaps clarify this 'using MIS 4 as an analogue we infer a larger Laurentide ice sheet compared to Eurasia'*

We have clarified this point in the text.

*Line 540: 'very large ice and or very cold water'. Does the deep water temperature reconstruction of Sosdian and Rosenthal offer any insights here?*

One has to be cautious with the deep-water temperature record of Sodian and Rosenthal (2009) because Mg/Ca was measured on the epibenthic *C. wuellerstorfi* that may be affected by variations in carbonate ion (see comment by Yu and Brocecker, 2010).

*Figure 1 - please ensure that the relevant permissions have been sought for the basemap of this figure, which is taken from Stokes and Clark 2001 (the only 'modifications' to this map are the additions of the site locations)*

Copyright clearance to reproduce Figure 1 has been sought and granted by Elsevier through the Copyright Clearance Center's RightsLink service.

[revised manuscript text omitted]

---

## Author Response (AR2)

We tried to address the many relevant issues raised by Reviewer 1 (Bailey) within the framework of the original paper, but agree that a more complete discussion is needed. Thus, as per the Editor's suggestion, we have added an additional section to the paper (4.6 Caveats) where we discuss the limitations of the current study and outstanding questions that will require additional millennially-resolved records from the North Atlantic.

The analytical precision for foraminifer isotope results has been added to the methods.

We have also corrected the age-depth table (Table S3) as two control points were omitted (0,0 for the top of the core and the top of the Mammoth Subchron that anchors the bottom of the core).